# Iterative Label Refinement Matters More than Preference Optimization under Weak Supervision

**Yaowen Ye** *
The University of Hong Kong
elwin@connect.hku.hk

**Cassidy Laidlaw** *
University of California, Berkeley
{cassidy_laidlaw, jsteinhardt}@berkeley.edu

**Jacob Steinhardt**

## Abstract

Language model (LM) post-training relies on two stages of human supervision: task demonstrations for supervised finetuning (SFT), followed by preference comparisons for reinforcement learning from human feedback (RLHF). As LMs become more capable, the tasks they are given become harder to supervise. Will post-training remain effective under unreliable supervision? To test this, we simulate unreliable demonstrations and comparison feedback using small LMs and time-constrained humans. We find that in the presence of unreliable supervision, SFT still retains some effectiveness, but DPO (a common RLHF algorithm) fails to improve the model beyond SFT. To address this, we propose *iterative label refinement* (ILR) as an alternative to RLHF. ILR improves the SFT data by using comparison feedback to decide whether human demonstrations should be replaced by model-generated alternatives, then retrains the model via SFT on the updated data. SFT+ILR outperforms SFT+DPO on several tasks with unreliable supervision (math, coding, and safe instruction-following). Our findings suggest that as LMs are used for complex tasks where human supervision is unreliable, RLHF may no longer be the best use of human comparison feedback; instead, it is better to direct feedback towards improving the training *data* rather than continually training the *model*. Our code and data are available at https://github.com/helloelwin/iterative-label-refinement.

## 1 Introduction

Language models (LMs) learn rich knowledge when pretrained on internet-scale corpora (Achiam et al., 2023; Dubey et al., 2024). To elicit their full capabilities and align them to human values, they are typically post-trained with two types of human supervision: task *demonstrations* for the initial supervised finetuning (SFT) stage, followed by preference *comparisons* used in reinforcement learning from human feedback (RLHF) (Ouyang et al., 2022; Bai et al., 2022) via algorithms like proximal policy optimization (PPO) (Schulman et al., 2017) or direct preference optimization (DPO) (Rafailov et al., 2024; Dubey et al., 2024).

As LMs continue to advance, they are trained to solve complex tasks that are difficult for humans to supervise (Amodei et al., 2016; Leike & Sutskever, 2023; Wen et al., 2024). For example, humans are often imperfect at coding, producing bugs that models like Github Copilot learn to imitate (Asare et al., 2023). Similarly, industrial-level ChatGPT training data rated as "flawless" has recently been found to contain flaws (McAleese et al., 2024). As task complexity increases, human supervision may become even less reliable. Thus, it is imperative to understand how both stages of current post-training pipelines (SFT + RLHF) perform under unreliable supervision.

Studying this is difficult because tasks where human supervision is unreliable are by nature difficult to obtain ground truth for. Ideally, we want tasks with known ground truth where we can also collect human-like unreliable data. We employ two approaches to simulate this unreliable supervision. First, we use smaller LMs that often make mistakes, in line with Burns et al. (2023) and Dubois et al. (2024). Second, we recruit human workers to label data under time constraints. For either of these two types, we can use it to simulate either *unreliable demonstrations* or *unreliable comparisons* (e.g. for

---

*Equal contribution

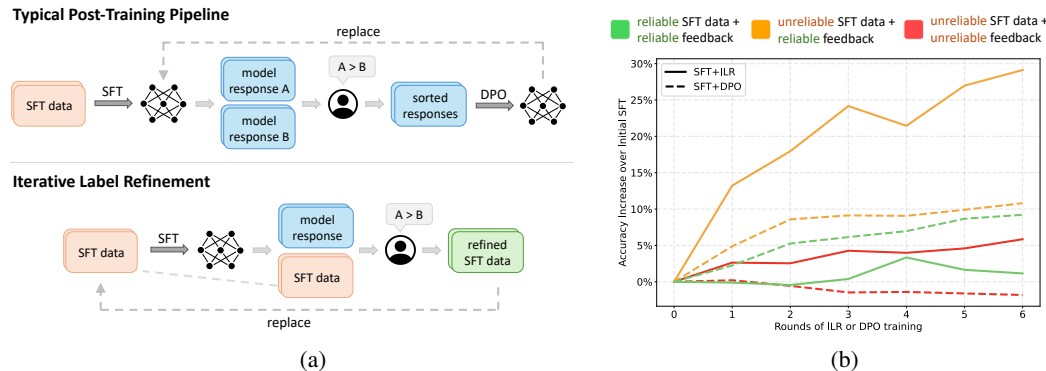

Figure 1: (a) In contrast to RLHF, which iteratively updates the SFT *model*, ILR directs comparison feedback towards improving the SFT *dataset*. (b) With reliable supervision, DPO effectively improves the SFT model and ILR is unnecessary. However, DPO becomes less effective with unreliable demonstrations and breaks down completely when comparison feedback is also unreliable. In contrast, ILR consistently improves on the initial SFT model with unreliable supervision, even in the most extreme case.

DPO). We thus simulate post-training by running SFT on unreliable demonstrations followed by DPO on unreliable comparisons.

We first find that SFT with unreliable demonstrations performs adequately, in that the learned SFT model is more reliable than the demonstrations themselves, a phenomenon termed weak-to-strong generalization (W2SG) that has been previously observed by Burns et al. (2023) in classification tasks. However, since these models inevitably learn to imitate errors in their unreliable training data, the resulting performance is still far from their full capability when trained on ground truth. Thus, ideally, the subsequent DPO stage should improve these SFT models.

Unfortunately, we find DPO breaks down under unreliable supervision. With unreliable demonstrations and comparisons, DPO offers little to no performance improvement on top of SFT (Figure 1b). Our experiments suggest that this may be due to the tendency of DPO to overoptimize, similar to what is observed in Gao et al. (2023). To prevent overoptimization, DPO implicitly penalizes KL divergence from the SFT model, and we find that a large penalty is necessary when feedback is unreliable. However, this large penalty prevents the large changes in the model that are necessary to correct the errors learned from the unreliable demonstrations during SFT (Section 4.1).

To address this, we propose *iterative label refinement* (ILR). In contrast to RLHF, which iteratively updates the SFT *model*, ILR updates the SFT *dataset* using the same type of comparison feedback. Specifically, ILR uses this feedback to compare a demonstration in the SFT dataset with an alternative one written by the SFT model; if the model-written demonstration is chosen, it replaces the original demonstration in the SFT data. This improves the SFT data by replacing low-quality or incorrect demonstrations with better model-written ones. Thus, when a new SFT model is trained on the updated dataset, it should perform better than the original SFT model, facilitating further improvements to the dataset via additional rounds of ILR (Figure 1a). Unlike DPO, we find that updating the SFT data via ILR leads to significant improvements from one round to the next (Figure 1b).

SFT+ILR improves over SFT+DPO on several tasks (math, coding, and safe instruction following) in our LM-based simulation. These results are further confirmed by our human study, where we recruit workers to provide demonstrations and comparison feedback under time constraints for the Alpaca (Taori et al., 2023) instruction-following task. In this setting with more realistic human errors, ILR continues to improve the initial unreliable human demonstrations and outperforms DPO.

In summary, our findings suggest that preference-based post-training like RLHF may no longer be the best use of comparison feedback as we train LMs to solve complex tasks where human supervision is unreliable. Instead, it is better to direct human feedback towards fixing errors in the unreliable training *data* rather than continually training the *model*.

## 2    RELATED WORK

**Scalable oversight and weak-to-strong generalization.** Scalable oversight aims to develop methods to supervise AI systems on tasks that are difficult for humans (Amodei et al., 2016; Bowman et al.,

2022). The primary focus in scalable oversight has been on designing human-AI collaboration mechanisms that enhance humans' ability to evaluate AI outputs more accurately (Christiano et al., 2018; Irving et al., 2018; Michael et al., 2023; Khan et al., 2024; Kenton et al., 2024) and with less cognitive burden (Saunders et al., 2022; McAleese et al., 2024; Kirchner et al., 2024). In contrast, recent work on W2SG (Burns et al., 2023) explores a complementary direction that develops learning algorithms to make models generalize correctly from weak supervision as if they were trained on higher-quality supervision or even ground truth. The SFT stage in our LM-based simulation employs a similar setting as the setup used for studying W2SG in Burns et al. (2023) but extended to text generation tasks; Burns et al. (2023) only consider classification tasks.

**Language model post-training.** Pretrained language models (PLMs) possess a rich understanding of language but require post-training to align their behavior with human preferences. This generally involves supervised finetuning (SFT) (Wei et al., 2021) on human-written demonstrations and reinforcement learning from human feedback (RLHF) (Ouyang et al., 2022; Bai et al., 2022). RLHF is typically done by algorithms like PPO (Schulman et al., 2017), which uses an explicit reward model, or DPO (Rafailov et al., 2024; Dubey et al., 2024) and its variants (Liu et al., 2023; Azar et al., 2024), which uses an implicit reward model. Recent work has also investigated the impact of noisy preference data on RLHF (Chowdhury et al., 2024; Fisch et al., 2024; Gao et al., 2024), while our work addresses a more challenging case where both SFT data and preference data are unreliable.

**Learning with imperfect supervision.** Research in weakly-supervised learning has focused on training models with noisy data (Reed et al., 2014; Rolnick et al., 2017; Song et al., 2022). Many works study classification with random label flips, proposing methods like data filtering and robust losses (Zhou, 2018; Frénay & Verleysen, 2013), while some also consider noise in structured prediction tasks like parsing and word alignment (Ganchev, 2010). Our setting differs as we study unreliable supervision from LMs or time-constrained humans for complex text-generation tasks. In this setting, errors tend to be *systematic*—arising from reasoning failures or social biases—rather than *random*, and so may not be well-addressed by traditional methods. Our cross-labeling framework shares conceptual similarities with co-training (Blum & Mitchell, 1998; Zhou et al., 2005) and co-teaching (Han et al., 2018), although these methods primarily focus on classification settings and rely on model confidence scores.

## 3 PROBLEM DEFINITION: POST-TRAINING WITH UNRELIABLE SUPERVISION

Training LMs to be useful assistants involves two stages: pre-training and post-training. During pre-training, the LM is trained to autoregressively predict a large corpus of text, typically taken from the internet and books (Radford et al., 2019; Brown et al., 2020). Afterwards, the pretrained language model (PLM) has acquired knowledge and skills from its pre-training data, but cannot yet be easily used for realistic applications. To elicit the PLM's capabilities for some class of tasks, post-training techniques are used (Wei et al., 2021; Dubey et al., 2024; Ouyang et al., 2022; Bai et al., 2022). We denote an LM as a conditional distribution $p(y \mid x)$ over responses $y$ given a prompt $x$.

**Standard post-training pipeline.** Post-training a PLM typically involves two stages. In the initial SFT stage, a PLM is trained on human-written task demonstrations. That is, given an SFT dataset $\mathcal{D}_{\text{SFT}} = (\mathcal{X}_{\text{SFT}}, \mathcal{Y}_{\text{SFT}})$ of prompts $\mathcal{X}_{\text{SFT}}$ and human-written responses $\mathcal{Y}_{\text{SFT}}$, the PLM is trained to minimize $\frac{1}{|\mathcal{D}_{\text{SFT}}|} \sum_{(x,y) \in (\mathcal{X}_{\text{SFT}}, \mathcal{Y}_{\text{SFT}})} -\log p(y \mid x)$. We call the result of this stage the SFT model, which has typically learned the correct input/output format for the given task and has partially elicited the PLM's capabilities.

Subsequently, to further improve performance, the SFT model undergoes RLHF via algorithms like PPO (Ouyang et al., 2022; Bai et al., 2022; Schulman et al., 2017), DPO (Rafailov et al., 2024; Dubey et al., 2024), or related methods. In this stage, a new dataset is constructed using a set of prompts $x \in \mathcal{X}_{\text{RLHF}}$ and response pairs $y, y'$ sampled from the SFT model. Each prompt and pair of responses are shown to a human annotator that picks the response they prefer, creating a dataset of triples $(x, y_+, y_-) \in \mathcal{D}_{\text{RLHF}}$, where $y_+$ is preferred to $y_-$. This dataset is used for preference optimization via methods like PPO, which trains a reward model from $\mathcal{D}_{\text{RLHF}}$ for online RL, or DPO, which uses an implicit reward model. In this work, we focus on DPO due to its computational efficiency and stability. We call the output of the this stage the SFT+DPO model.

**Simulating unreliable human supervision** To study how current post-training pipelines perform under unreliable supervision, we need tasks with known ground truth so that we can evaluate

performance, along with realistic forms of unreliable supervision. We employ two approaches to simulate unreliable supervision: small language models (Section 4 and 5) and time-constrained humans (Section 6). Most of our experiments focus on the small-LM case, in line with Burns et al. (2023) and Dubois et al. (2024), and we then verify the results on our instruction-following dataset using time-constrained human annotators.

For both forms of supervision, we need to simulate both *unreliable demonstrations* and *unreliable comparisons*. For humans, we do this simply by querying them. For small LMs, we follow the procedure below (more details in Appendix B):

1. *Unreliable task demonstrations*: We finetune a small PLM $\tilde{p}$ on ground-truth demonstrations and use it to generate responses for a held-out set of prompts as unreliable demonstrations.

2. *Unreliable comparison feedback*: We finetune a classification model $\tilde{q}$ to select the better response when given two responses $y_1, y_2$ and a prompt $x$, where $\tilde{q}(y_1 \succ y_2 \mid x)$ denotes the probability that $y_1$ is preferred to $y_2$. Note that $\tilde{q}$ is based on the same PLM as $\tilde{p}$ but with an additional classification head. Each element of $\tilde{q}$'s training data consists of a prompt and two responses: one response is from the ground-truth demonstrations used to train $\tilde{p}$ and the other is a lower-quality output generated by intermediate checkpoints of $\tilde{p}$. $\tilde{q}$ is trained to choose the ground truth using a standard binary classification loss.

**SFT+DPO with unreliable supervision.** We start by finetuning a larger PLM on an unreliable SFT dataset $\widetilde{\mathcal{D}}_{\text{SFT}} = (\mathcal{X}, \widetilde{\mathcal{Y}})$ to obtain an initial SFT model $\widehat{p}_{\text{SFT}}$, where $\widetilde{\mathcal{Y}}$ is a set of unreliable demonstrations given by $\tilde{p}$ or time-constrained humans. We then perform $K$ rounds of DPO training. In each round $k$, we: sample completions $y_1, y_2$ from the current SFT+DPO model $\widehat{p}_{\text{SFT+DPO}}^{k-1}$ for each prompt $x \in \mathcal{X}$ (where $\widehat{p}_{\text{SFT+DPO}}^0 \equiv \widehat{p}_{\text{SFT}}$); collect unreliable comparisons using either $\tilde{q}$ or time-constrained humans to construct an unreliable preference dataset $\widetilde{\mathcal{D}}_{\text{RLHF}}^k$; and, use this to train $\widehat{p}_{\text{SFT+DPO}}^{k-1}$ via DPO, resulting in $\widehat{p}_{\text{SFT+DPO}}^k$. The final model after $K$ rounds is denoted as $\widehat{p}_{\text{SFT+DPO}}^K$ or simply $\widehat{p}_{\text{SFT+DPO}}$. Further implementation and hyperparameter details can be found in Appendix C.

## 3.1 TASKS AND MODELS

**Datasets.** We test SFT+DPO with unreliable feedback on three text generation tasks: mathematical problem-solving using GSM8K (Cobbe et al., 2021), SQL code generation with BIRD (Li et al., 2024), and safe instruction following with SaferPaca (Bianchi et al., 2023), which is a mix of the Alpaca dataset (Taori et al., 2023) and refusal demonstrations to unsafe instructions. All datasets are formatted as question-answer pairs following the same template.

**Models.** We use three open-source PLMs of varying sizes in our experiments: Gemma 2B (Team et al., 2024), Mistral 7B (Jiang et al., 2023), and Meta Llama 3 70B (Dubey et al., 2024). Our LM-simulated experiments include four settings that use a smaller model to supervise a larger model. In our simulation with time-constrained humans, we only experiment with the largest 70B model.

**Evaluation metrics.** Each dataset requires a specific approach to evaluate model outputs and determine performance. For GSM8K, we parse the numerical answer following "####" at the end of each response and compute exact match accuracy by comparing it with the ground truth. For BIRD, we measure execution accuracy by running the generated code on corresponding test databases, following Li et al. (2024). For SaferPaca, we follow Li et al. (2023) and use GPT-4o (OpenAI, 2024) to compute win rate against reference answers.

Further implementation details on datasets, evaluation metrics, model architectures, training and inference configurations are provided in Appendix A, B, and C.

## 4 DPO IS INEFFECTIVE WITH UNRELIABLE SUPERVISION

In this section, we present results of SFT+DPO under LM-simulated unreliable supervision. We find that SFT still retains some effectiveness, but DPO fails to improve the SFT model. We further show that the failure of DPO appears to be caused by its tendency to overoptimize given unreliable comparison feedback, which motivates our alternative approach introduced in the next section.

**SFT shows limited robustness to unreliable demonstrations.** We first investigate the effectiveness of SFT under LM-simulated unreliable supervision. When finetuned on $\widetilde{\mathcal{D}}_{\text{SFT}}$ with unreliable demon-

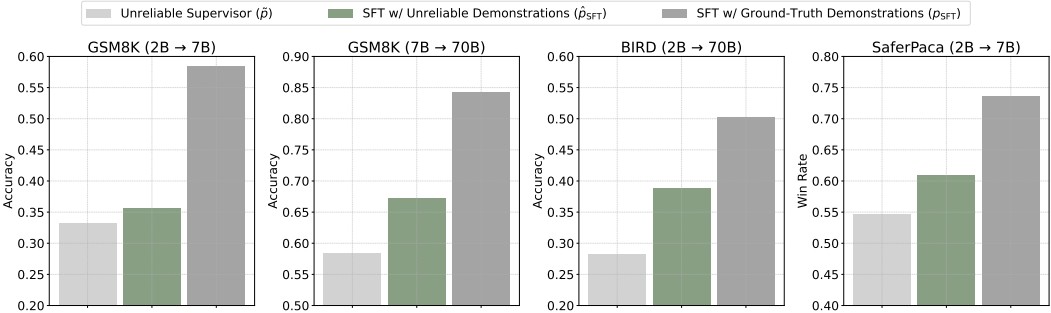

Figure 2: Models finetuned on unreliable demonstrations generated by a smaller supervisor model show higher accuracy than the supervisor, demonstrating SFT's robustness to unreliable supervision. However, SFT models' full capability is only partially recovered with unreliable demonstrations. A → B denotes using demonstrations generated by the model with size A to finetune the model with size B.

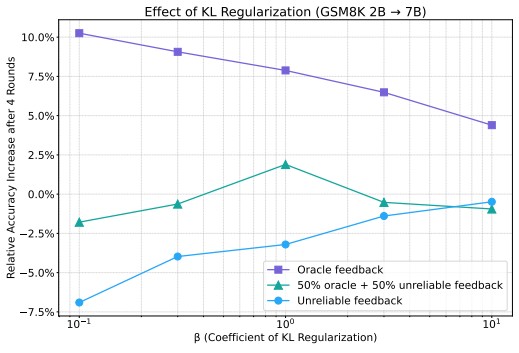

(a) DPO effectively improves SFT models with oracle comparisons, especially with weak regularization. However, with unreliable feedback, improvements are limited to a narrow range of regularization strengths because weak regularization causes overoptimization.

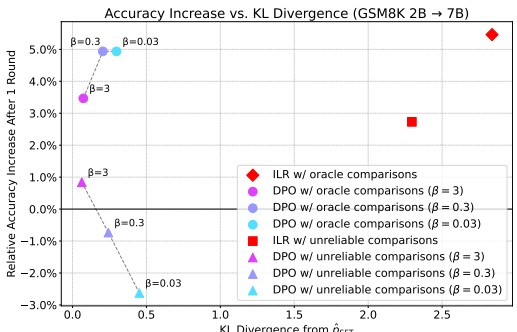

(b) Strong regularization in DPO limits useful model updates, while small regularization leads to overoptimization of unreliable feedback. In contrast, ILR facilitates large model updates, allowing for faster improvement and efficient use of comparison feedback.

Figure 3: DPO struggles to avoid overoptimization of noisy preferences while also updating the initial suboptimal SFT model sufficiently to improve performance.

strations generated by $\tilde{p}$, the larger model $\widehat{p}_{\text{SFT}}$ consistently outperforms $\tilde{p}$ across all tasks, showing SFT's robustness to imperfect demonstrations (Figure 2).

However, models finetuned on unreliable demonstrations inevitably learn to imitate errors in them. When applying SFT to the same PLM using *ground-truth* demonstrations, which we denote $p_{\text{SFT}}$, we find that it significantly outperforms $\widehat{p}_{\text{SFT}}$ (Figure 2). This shows that SFT on unreliable demonstrations only partially recovers a model's full capability, leaving a large performance gap between $p_{\text{SFT}}$ and $\widehat{p}_{\text{SFT}}$. Thus, ideally, the subsequent DPO stage should further improve upon $\widehat{p}_{\text{SFT}}$.

**DPO does not improve over SFT with unreliable comparisons.** Contrary to DPO's past success (Rafailov et al., 2024; Dubey et al., 2024), we find that it consistently fails to improve performance over SFT when both demonstrations and comparison feedback are unreliable (Figure 4). That is, when we further finetune $\widehat{p}_{\text{SFT}}$ via multiple rounds of DPO on comparison feedback given by $\tilde{q}$, the resultant model $\widehat{p}_{\text{SFT+DPO}}$ shows little to no performance gain (or performance even declines).

To investigate the degree to which this is due to the unreliable comparison feedback versus the unreliable SFT data, we run DPO in two other settings on the GSM8K task. In the first, we initialize training with $\widehat{p}_{\text{SFT}}$ but use an oracle evaluator for preference comparisons that always selects the better answer by comparing model outputs to ground truth. In the second, we again use the oracle evaluator, but start from an SFT model trained on a filtered subset of $\widetilde{\mathcal{D}}_{\text{SFT}}$ that only contains correct demonstrations. As shown in Figure 1b, DPO with the oracle evaluator does manage to improve over the starting SFT model in both cases. These results reveal that DPO's effectiveness heavily relies on high-quality supervision.

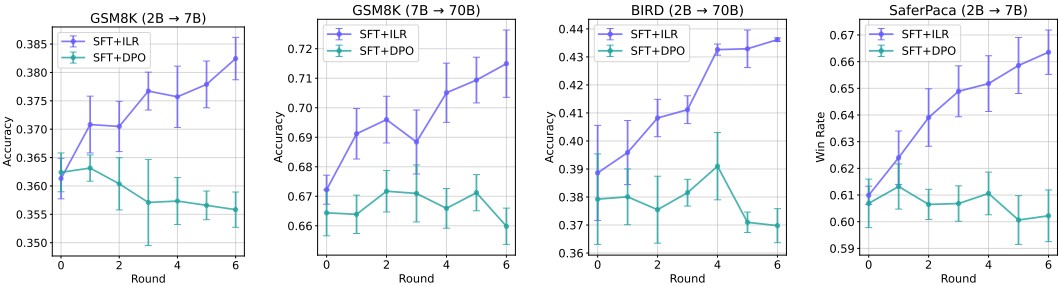

Figure 4: ILR consistently provides more improvements for the SFT model than DPO in four settings with LM-simulated unreliable demonstrations and comparison feedback. Round 0 represents accuracy of $\widehat{p}_{\text{SFT}}$.

## 4.1 DPO with Unreliable Feedback Exhibits Overoptimization

We hypothesize that DPO's failure under unreliable supervision stems from the problem of overoptimization that preference-based learning methods often suffer from (Gao et al., 2023). Optimizing an explicit reward function (with PPO) or an implicit one (with DPO) based on preference feedback can initially lead to an increase in performance but eventually causes a decline as the reward function diverges from the true objective. To combat this, both PPO and DPO regularize based on the KL divergence between the initial and updated policies.

We find that the optimal amount of regularization for DPO, which is controlled by the hyperparameter $\beta$, depends strongly on the reliability of comparison feedback. This dependency creates a critical trade-off: stronger regularization is needed to prevent overoptimization with unreliable feedback, but this constrains DPO's ability to improve upon the suboptimal SFT model. To demonstrate this, we measure DPO's relative improvement over SFT under varying feedback quality levels and KL regularization strengths (Figure 3). Full results for each training epoch are presented in Figure 8.

**With unreliable comparison feedback, DPO is especially prone to overoptimization and needs very strong regularization.** In Figure 3a, we consider three levels of feedback quality: unreliable feedback from $\tilde{q}$, mixed feedback (50% $\tilde{q}$, 50% oracle), and pure oracle feedback. With oracle feedback, smaller $\beta$ values consistently yield better performance. However, as feedback quality decreases, smaller $\beta$ values lead to worse performance while only a suitably large $\beta$ enables positive improvements. This suggests that preference optimization under unreliable feedback needs heavy regularization to prevent overoptimization.

**Strong regularization limits useful model updates during DPO training.** Figure 3b demonstrates that larger divergence correlates with higher accuracy increases in the case of oracle feedback (colored circles in upper left). This indicates that setting small $\beta$ to allow substantial model updates is crucial for improving upon the initial suboptimal SFT model. However, with unreliable feedback, this is not possible: as we observed above, using insufficient regularization with unreliable comparisons leads to overoptimization.

These two observations suggest a tension between preventing overoptimization from unreliable comparison feedback and making large enough model updates. Since SFT on unreliable demonstrations produces a model $\widehat{p}_{\text{SFT}}$ that is suboptimal, we need large updates to $\widehat{p}_{\text{SFT}}$ to recover performance close to $p_{\text{SFT}}$. However, with unreliable comparison feedback, we need significant regularization (*i.e.*, large $\beta$) for DPO, which prevents the large updates that are needed. Given these issues, we seek an alternative algorithm that better leverages comparison feedback to improve over the SFT model.

## 5 Iterative Label Refinement

To overcome the limitations of preference optimization discussed in Section 4.1, our method needs to be robust to unreliable comparisons while still allowing large updates from the SFT model. To achieve this, we propose *iterative label refinement* (ILR). In contrast to RLHF methods, which iteratively update the SFT *model*, ILR focuses on improving the unreliable demonstrations in the initial SFT *dataset*. It uses comparison feedback to decide whether the initial SFT data should be replaced by model-generated alternatives, and then retrains the model from scratch via SFT on the

new data (Figure 1a). As we will show, ILR avoids the regularization dilemma discussed in Section 4.1 that impedes DPO with unreliable feedback. In ILR, retraining the model from scratch at each iteration allows for large changes to the SFT model but avoids overoptimization. We introduce the detailed methodology of ILR in Section 5.1, and demonstrate that it outperforms DPO both in the LM-simulated setting (Section 5.2) and with time-constrained humans (Section 6).

## 5.1 METHODOLOGY

ILR consists of several iterations; during each iteration, the SFT dataset is improved by replacing some of the demonstrations. The first step of each iteration is to gather proposed demonstrations that may replace unreliable demonstrations in the current SFT data. These proposals are generated using models trained via SFT on the current dataset. We showed in Section 4 that SFT models often outperform their unreliable training data. Thus, by replacing some of the existing dataset with the improved output of the SFT model, the overall quality and accuracy of the dataset should increase.

However, our findings in Section 4 only show that SFT models outperform their training data on *held-out* prompts; if an SFT model is tested on a train prompt, it is likely to output responses that contain mistakes imitated or memorized from the training data. Thus, if we use a model trained on the entire current SFT dataset to generate proposals, those proposed responses are less likely to improve over the current responses in the dataset. To ensure that the proposed replacements are generated on held-out prompts, we implement ILR by training *two* SFT models on different halves of the SFT data; then, these models cross-label the half they were not trained on. This makes it more likely the model-generated responses are different and better than the existing responses, enabling improvement of the dataset.

Once proposal responses are generated, we selectively update the SFT dataset with them by leveraging comparison feedback. We ask the annotator (*i.e.*, the small LM $\tilde{q}$ or time-constrained humans) to compare an existing response in $\widetilde{\mathcal{D}}_{\text{SFT}}$ with a model-generated proposal for the same prompt. If the proposal is preferred by the annotator, it replaces the original response. After these comparisons are complete, a new SFT model is trained on the updated dataset. As long as the annotator chooses better responses more than half the time, the overall accuracy of the SFT data will increase, enabling the new SFT model to outperform the initial one.

Formally, each iteration of ILR consists of the following steps:

1. *Data splitting*: We start with a dataset of task demonstrations $\mathcal{D}_k = (\mathcal{X}, \tilde{\mathcal{Y}}_k)$, where $k$ is the current iteration and $\mathcal{D}_0 \equiv \widetilde{\mathcal{D}}_{\text{SFT}}$. The dataset $\mathcal{D}_k$ is evenly split into two disjoint subsets, $\mathcal{D}_k^1 = (\mathcal{X}^1, \mathcal{Y}_k^1)$ and $\mathcal{D}_k^2 = (\mathcal{X}^2, \mathcal{Y}_k^2)$.

2. *Model training*: Two models, $\widehat{p}_{\text{SFT}}^1$ and $\widehat{p}_{\text{SFT}}^2$, are finetuned on $\mathcal{D}_k^1$ and $\mathcal{D}_k^2$ respectively.

3. *Cross-labeling*: Each model generates label proposals for the subset it wasn't trained on, *i.e.*, we sample $\mathcal{Z}_k^2 \sim \widehat{p}_{\text{SFT}}^1(\cdot \mid \mathcal{X}^2)$ and $\mathcal{Z}_k^1 \sim \widehat{p}_{\text{SFT}}^2(\cdot \mid \mathcal{X}^1)$.

4. *Proposal evaluation*: The unreliable supervisor provides comparison feedback on the new proposal $z_i$ and original label $\tilde{y}_i$ for a prompt $x_i$ to decide which of them is preferred; this feedback determines whether $\tilde{y}_i$ should be replaced by $z_i$.

5. *Label updating*: Based on the comparisons in step 4, we form an updated dataset $\mathcal{D}_{k+1} = (\mathcal{X}, \tilde{\mathcal{Y}}_{k+1})$, where each element of $\tilde{\mathcal{Y}}_{k+1}$ is either the accepted new proposal or the retained original label. We control the label refinement speed by setting a hyperparameter $\alpha \in (0, 1]$, allowing at most $\alpha|\mathcal{X}|$ labels to be updated in each iteration. When more than $\alpha|\mathcal{X}|$ proposals are accepted in step 4, we choose the ones with higher evaluation confidence, determined by either $\tilde{q}$'s log probability or human annotators' self-reported confidence. In all experiments we set $\alpha = 0.15$. Further analysis of $\alpha$ is presented in Appendix E.3.

At the end of each iteration, we finetune a new model starting from the base PLM of $\widehat{p}_{\text{SFT}}$ using the entire refined dataset $\mathcal{D}_K$, resulting in the SFT+ILR model $\widehat{p}_{\text{SFT+ILR}}^k$. This process is repeated for $K$ iterations and the final model is $\widehat{p}_{\text{SFT+ILR}}^K$.

In the LM-simulated setting, we only consider proposals that are sufficiently different from the original labels during step 4. We determine a proposal is different enough if it has a different final

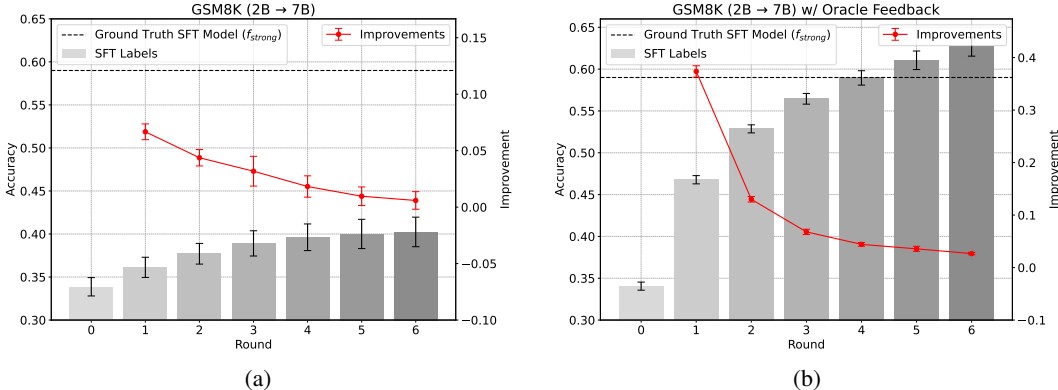

Figure 5: SFT label accuracy increases across rounds during ILR. With high-quality feedback, ILR can lead to labels that approach or even surpass the accuracy of the model trained on ground truth ($p_{\text{SFT}}$).

answer (GSM8K), different execution result (BIRD), or large embedding distance (SaferPaca). We present the pseudo-code (Algorithm 1) and more implementation details of ILR in Appendix C.

## 5.2 LM-SIMULATED EXPERIMENT RESULTS

**SFT+ILR leverages unreliable supervision better than SFT+DPO.** We compare SFT+ILR and SFT+DPO across four settings with LM-simulated unreliable demonstrations and comparison feedback (further validation using time-constrained human data is presented in Section 6). As shown in Figure 4, SFT+ILR consistently outperforms SFT+DPO in all scenarios under unreliable supervision. In GSM8K, ILR shows more significant improvements as model size increases from 7B to 70B, suggesting that it benefits from model scaling and may remain effective for future models that are even more capable.

**Comparison feedback is effective for improving demonstration quality.** ILR relies on refining SFT data using comparison feedback, which is often easier to obtain than high-quality demonstrations, especially for complex tasks that humans struggle with. Figure 5a shows that unreliable comparison feedback guides this refinement process effectively: the accuracy of the SFT data steadily increases with more rounds of ILR. As suggested, this is likely because evaluation of AI output is typically easier than the demonstration of an ideal output (Leike et al., 2018), allowing even imperfect feedback to contribute to meaningful improvements in the SFT data. With reliable feedback (Figure 5b), the SFT demonstrations can be improved even further, approaching or even surpassing the accuracy of the model $\widehat{p}_{\text{SFT}}$ trained on ground truth. This highlights ILR's potential when combined with other scalable oversight techniques that enhance humans' evaluation capability.

**ILR enables larger model updates and more efficient use of comparison feedback.** As illustrated in Figure 3b, models trained with ILR exhibit significantly higher KL divergence from the initial SFT model compared to those trained with DPO even after a single round. Moreover, when comparison feedback is reliable while SFT data is not, ILR shows much higher efficiency than DPO, using the same amount of feedback to enable larger improvement over the initial SFT model. These can be attributed to ILR's direct modification of the unreliable SFT data, which fundamentally alters models' training dynamics during SFT. By doing so, ILR allows each subsequent model to learn from new data independently and from scratch, without inheriting the errors of previous models, unlike the continual preference-based training seen in DPO.

**Supervision for refinement is necessary.** Since the SFT models can generate proposal responses that are more accurate than their training data, it appears that one can simply replace the initial SFT data with new proposals without using any comparison feedback. To understand the importance of comparison feedback in ILR's refinement process, we compare it to a naive approach that directly replaces an $\alpha$ fraction of the original labels with new proposals in each round. As shown in Figure 9 (Appendix E.2), this naive method leads to performance degradation, showing that supervision in the refinement process, even if unreliable, is necessary. It could be that training on model outputs without any curation leads to model collapse, a phenomenon observed when training generative models with synthetic data (Shumailov et al., 2023; Ren et al., 2024; Gerstgrasser et al., 2024).

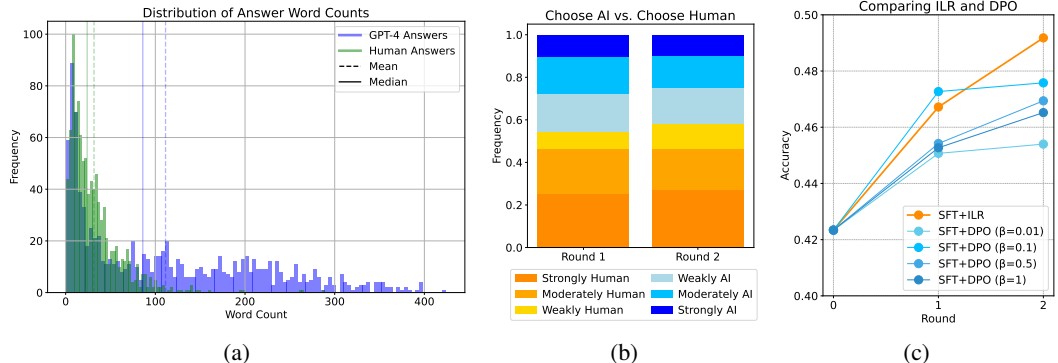

(a)                                    (b)                                    (c)

Figure 6: (a) Distribution of word count of human written demonstrations compared to GPT4-generated answers collected by (Peng et al., 2023). (b) In both ILR rounds, participants are at least moderately confident in choosing newly proposed labels over original human demonstrations 20% of the time, but they tend to update less in later rounds. Participants also show higher overall confidence when choosing human demonstrations. These suggest that W2SG is not effective enough to consistently produce better outputs or that AI outputs are becoming more complex for time-constrained participants to evaluate. (In the second round, we only count preference pairs with at least one human-written label when calculating the frequencies) (c) ILR enables more improvements on top of the initial SFT model, while DPO plateaus similar to what is observed in LM-simulated settings.

| Question | Human Demonstration | Chosen AI output |
|---|---|---|
| Below is an instruction that describes a task... Instruction: Explain the concept of random forest. | A random forest is a forest, usually in nature, that is not accounted for with maps or other documentation. | A random forest is a machine learning method for predication or classification using multiple decision trees... |

Figure 7: Example of lable refinement in ILR: When instructed to explain "random forest," an inaccurate human demonstration misinterpreted it as a natural forest. Human workers then chose to replace this with a more accurate model-generated explanation, improving the SFT dataset quality.

# 6    TIME-CONSTRAINED HUMAN STUDY

To validate our findings from LM-simulated settings, we conduct a human study where we recruit workers through CloudResearch Connect[1] to provide both task demonstrations and comparison feedback for an instruction-following task. Although the task is not beyond human capability, we impose time constraints to create annotator noise and obtain unreliable data; this simulates the challenges that may arise as humans supervise AI on complex tasks that are difficult for humans themselves.

**Task and data collection.** For simplicity, we focus on the Alpaca instruction-following dataset (Taori et al., 2023; Peng et al., 2023) without unsafe instructions. We evaluate models using the complete test set of AlpacaEval (Li et al., 2023) and GPT-4o as a judge, similar to evaluation in our SaferPaca setting. We collect human demonstrations for 1,000 randomly sampled instructions, with workers instructed to spend 1-2 minutes to write each response. Each worker is assigned 15 questions, plus 3 attention checks used to filter out low-quality responses. As shown in Figure 6a, human demonstrations are generally shorter than GPT-4-generated responses, suggesting suboptimal quality due to time constraints. For the comparison feedback used in DPO and ILR, we instruct workers to spend 30 seconds to 1 minute per question, with 50 questions assigned per worker and 3 additional questions for quality checks. We collect 1,000 comparisons for each algorithm in each round and conduct two rounds of both DPO ($\beta = 0.1$) and ILR ($\alpha = 0.1$). We also use the comparison data collected for DPO with $\beta = 0.1$ to run DPO with several other $\beta$ values. More details of our human study setup are provided in Appendix D.

---

[1]https://connect.cloudresearch.com

**ILR outperforms DPO with time-constrained human supervision.** With time-constrained unreliable human supervision, ILR demonstrates fast and consistent improvement compared to DPO (Figure 6c). While DPO with strong regularization (large $\beta$) improves slowly, DPO with less regularization (small $\beta$) improves quickly in the beginning but then plateaus in performance. We hypothesize that DPO may be overoptimizing with insufficient regularization, similarly to our LM-simulated experiments. Notably, the results of ILR and DPO with the collected time-constrained human data are most similar to LM-simulated scenarios with unreliable demonstrations but *reliable* comparison feedback, as seen in Figure 3. This suggests that comparing Alpaca outputs remains relatively straightforward for humans even under time pressure. Future work could explore more complex tasks or even shorter time constraints to better simulate more unreliable human feedback.

**ILR effectively improves SFT data because models finetuned on unreliable demonstrations can surpass human supervisors.** We find humans are at least moderately confident in choosing model-generated outputs over original human demonstrations for more than 20% of the questions in both rounds of ILR (Figure 6b). This further confirms the W2SG phenomenon, where models trained on unreliable supervision can surpass their supervisor. For example, Figure 7 demonstrates that a participant chose to replace human-written misinterpretation of "random forest" with a more correct and detailed model-generated explanation.

However, the effect of ILR may decrease over rounds because participants tend to select fewer AI-generated responses in later rounds (Figure 6b), which may result in fewer opportunities for label refinement. This could be due to two reasons. First, participants may become less confident in choosing AI outputs because in later rounds these outputs are more likely to surpass human level in complexity. Second, W2SG may not be effective enough to consistently produce strictly better AI outputs. To address these challenges, future work could explore better refinement mechanisms that combine human demonstrations with AI outputs, potentially allowing for more nuanced refinement in cases where both human and AI responses are partially correct in complementary ways.

## 7 DISCUSSION

In this work, we study the effectiveness of a typical SFT+DPO pipeline for language model post-training under unreliable supervision, as simulated by small LMs and time-limited humans. We show that in this setting, unlike with reliable feedback, SFT+DPO fails to improve upon SFT. Our analysis suggests that DPO struggles to avoid overoptimization with unreliable comparison feedback and requires heavy regularization, preventing it from correcting significant errors in the SFT model. To address this, we propose ILR to redirect comparison feedback towards improving unreliable demonstrations in the SFT dataset. ILR enables larger model updates without the overoptimization risks in preference optimization and consistently outperforms DPO under unreliable supervision.

**Limitations.** Our focus on RLHF via DPO may not fully capture the nuances of RLHF pipelines that use reward modeling and PPO. Future research should verify whether our findings generalize to these more complex RLHF setups. Also, our human study using time-constrained data collection may not perfectly simulate human errors that arise from more realistic capability constraints. Exploring more challenging tasks such as competition math or coding could potentially provide a closer analogy to supervising AI on tasks that humans truly struggle with.

**Future work.** Our findings open several exciting directions for future research. For example, one could explore hybrid approaches that combine ILR with RLHF. Applying PPO or DPO after several rounds of ILR could potentially yield more improvements, since the problem of imitating errors in unreliable demonstrations is relieved after improving the SFT data with ILR. Additionally, investigating more sophisticated label refinement strategies, such as synthesizing human demonstrations with model-generated proposals or implementing ensemble techniques through more than two cross-labeling splits, could potentially further enhance the effectiveness of ILR.

**Broader impact.** Our findings suggest a potential paradigm shift in how we approach human oversight of AI systems under unreliable supervision. We show that the canonical RLHF post-training pipeline may no longer be the best use of human comparison feedback. Instead, we need methods like ILR that leverage model outputs to improve and learn from unreliable human supervision.

ACKNOWLEDGMENTS

Thanks to Alex Shypula, Eli Bronstein, Meena Jagadeesan, Ruiqi Zhong, Xinyan Hu, and members of JS's group for helpful feedback, and to Tianyi Qiu, Xiaoyuan Zhu, Yitao Liu, Meitong Liu, Han Li for valuable discussions. This work is supported by the Open Philanthropy Action Fund through funding provided to JS. CL is supported by an Open Philanthropy AI Fellowship. Additional thanks to Yuchen for her personal support and encouragement.

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

# A    TASKS AND EVALUATION

## A.1    DATASETS

**GSM8K** (Cobbe et al., 2021): This dataset contains math word problems widely used to evaluate language models' mathematical problem solving capabilities. In total, there are 8.5K question-answer pairs for training and 3K for testing.

**BIRD** (Li et al., 2024): In this task, models are required to generate an SQL query that answer a question of interest, given context of database descriptions. We filter out questions with token lengths greater than 1024 and obtained 8.5K examples for training and 1.4K for testing.

**SaferPaca** (Bianchi et al., 2023): This dataset is a mixture of instruction following questions from (Taori et al., 2023) and demonstrations of refusals to unsafe instructions. It requires a model to serve as an assistant that follows human users' instructions while generating responsible responses and rejecting harmful prompts. For efficiency, we use a subset that contains 9500 general instructions with responses labeled by GPT-4 (collected by Peng et al. (2023)) and 500 safety-related instructions with the original labels in (Bianchi et al., 2023). The test set we use contains 300 questions and reference answers taken from AlpacaEval (Li et al., 2023), alongside 278 safety questions and simple rejections ("Sorry, I cannot help with that.") of the I-MaliciousInstructions and I-CoNA datasets used in (Bianchi et al., 2023).

We format all datasets into question-answer pairs following the same template:

> **Prompt template for all datasets**
>
> ```
> USER:\n{question}\n\nASSISTANT:{answer}\n
> ```

## A.2    EVALUATION

**Sampling.** For all tasks and models, we sample one response for each prompt using vLLM (Kwon et al., 2023) with beam search enabled and beam size set to 4.

**Metrics.** For GSM8K, we simply parse the final numerical answer and compute exact match accuracy. For BIRD, we download all databases and execute the generated code on them to compute execution accuracy, following (Li et al., 2024). We ensure all models' generated code is executed on the same machine for fair comparison. For SaferPaca, we adopt the CoT prompt (shown below) from (Li et al., 2023) and use GPT-4o (OpenAI, 2024) to compute model answers' win rates against the reference answers. Specifically, we compute GPT-4o's mean probability of generating a token that chooses the model's answer over the reference answer.

> **Prompt template for GPT-4o evaluation**
>
> ```
> Select the output A or B that best matches the given instruction.
> Choose your preferred output, which can be subjective. Your answer
> should first contain a concise explanation and then it should end
> with ONLY the token: 'A' or 'B' (no dot, no backticks, no new line,
> no quotes, no 'it depends', no 'both' ...), because we will use
> 'output[-1]' to extract the best output.
>
> Here's an example:
>
> # Example:
> ## Instruction:
> Give a description of the following job: "ophthalmologist"
>
> ## Output A:
> An ophthalmologist is a medical doctor who specializes in the
> diagnosis and treatment of eye diseases and conditions.
>
> ## Output B:
> ```

```
An ophthalmologist is a medical doctor who pokes and prods at your
eyes while asking you to read letters from a chart.

## Concise explanation followed by 'A' or 'B'

### Concise explanation
A provides a comprehensive and accurate description of the job of an
ophthalmologist. In contrast, B is more of a joke.

### Which is best, 'A' or 'B'?
A

# Task:
Now is the real task.

## Instruction:
{instruction}

## Output 'A':
{A}

## Output 'B':
{B}

## Concise explanation followed by 'A' or 'B'

### Concise explanation
```

## B  LM-BASED SIMULATION DETAILS

For GSM8K, we experiment both with using the 2B model to supervise the 7B model, and using the 7B model to supervise the 70B model. For SaferPaca and BIRD, we consider 2B supervising 7B and 2B supervising 70B, respectively, to balance limited compute while maintaining sufficient performance gaps between models.

In each task, we split the training set into two halves. We use the first half as ground truth to train $\tilde{p}$ and $\tilde{q}$, and then use $\tilde{p}$ to generate labels for the second half or use $\tilde{q}$ to generate comparison feedback for answers to questions in the second half. This is similar to the setup in (Burns et al., 2023), despite that we are operating on text generation tasks with both task demonstrations and comparison feedback.

To collect training data for $\tilde{q}$, we evenly save 10 intermediate checkpoints when training $\tilde{p}$ and use them to sample answers for questions in their training set. Then, we pair these low-quality answers with ground truth answers, and train $\tilde{q}$ to select the ground truth with a standard binary classification loss. In SaferPaca, we balance refusals and acceptances (*i.e.*, following the instruction) in the training data to avoid class bias in $\tilde{q}$. This is done by first training two $\tilde{p}$ on safe and unsafe instructions in SaferPaca respectively and sampling answers from them for each instruction. In this way, we obtain both acceptance and refusal responses for each instruction, and then pair them up with the ground truth as $\tilde{q}$'s balanced training data.

We use the following template to compose two answers and the question into a prompt for $\tilde{q}$:

**Prompt template for $\tilde{q}$**

```
QUESTION:
{{question}}

ANSWER (A):
{{answer_a}}

ANSWER (B):
```

Table 1: Training Hyperparameter for Each Dataset

| Dataset | Epoch | Batch Size | Max Answer Token |
|---------|-------|------------|------------------|
| GSM8K | 2 | 32 | 256 |
| BIRD | 2 | 32 | 256 |
| SaferPaca | 4 | 32 | 512 |

---

**Algorithm 1:** Iterative Label Refinement (ILR)

---

**Input** : Initial SFT dataset $\mathcal{D}_0 = (\mathcal{X}, \tilde{\mathcal{Y}}_0)$
  Comparison feedback annotator $\tilde{q}$
  Maximum iterations $K$
  Refinement rate $\alpha$
**Output** : Refined dataset $\mathcal{D}_K$
  Final model $\hat{p}_{\text{SFT+ILR}}^K$

1 **for** $k = 0$ **to** $K - 1$ **do**
2    Split $\mathcal{D}_k$ into two disjoint subsets:
3    $\mathcal{D}_k^1 = (\mathcal{X}^1, \tilde{\mathcal{Y}}_k^1)$ and $\mathcal{D}_k^2 = (\mathcal{X}^2, \tilde{\mathcal{Y}}_k^2)$, where $\mathcal{X}^1 \cup \mathcal{X}^2 = \mathcal{X}$
4    Train models $\hat{p}_{\text{SFT}}^1$ on $\mathcal{D}_k^1$ and $\hat{p}_{\text{SFT}}^2$ on $\mathcal{D}_k^2$
5    **foreach** $x_i \in \mathcal{X}^1$ **do**
6      Sample one proposal $z_i \sim \hat{p}_{\text{SFT}}^2(\cdot \mid x_i)$
7    **foreach** $x_i \in \mathcal{X}^2$ **do**
8      Sample one proposal $z_i \sim \hat{p}_{\text{SFT}}^1(\cdot \mid x_i)$
9    **foreach** $x_i \in \mathcal{X}$ **do**
10      Collect comparison feedback for $(z_i, \tilde{y}_{k,i})$ using $\tilde{q}$
11      Proposal $z_i$ is accepted if the feedback is $z_i \succ \tilde{y}_{k,i}$
12    Let $A$ be the set of indices where $z_i$ is accepted
13    Limit $|A| \leq \alpha|\mathcal{X}|$ by selecting proposals with highest confidence
14    Update labels:
15    $\tilde{y}_{k+1,i} = \begin{cases} z_i & \text{if } i \in A, \\ \tilde{y}_{k,i} & \text{otherwise} \end{cases}$
16    Form the updated dataset $\mathcal{D}_{k+1} = (\mathcal{X}, \tilde{\mathcal{Y}}_{k+1})$
17    Train model $\hat{p}_{\text{SFT+ILR}}^{k+1}$ via SFT on $\mathcal{D}_{k+1}$
18 **return** $\mathcal{D}_K, \hat{p}_{\text{SFT+ILR}}[K]$

---

```
{{answer_b}}

You gave the answer (A) to the question. Do you accept to
change it to answer (B) proposed by another model?
```

Note that the prompt states an order of "your answer" and "answer proposed by another model" due to historical reasons. In practice, we randomize the two answers' order to avoide position bias.

## C  IMPLEMENTATION DETAILS OF SFT, DPO, AND ILR

In all our experiments, we apply Low-Rank Adaptation (LoRA) (Hu et al., 2021) for efficient model training. We set $r = 64$ and $\alpha = 128$ for all models.

**SFT.** For each task, we use a consistent setting of epoch, batch size, and max answer token for all models (Table 1). We use set learning rate to $5e^{-4}$ for Gemma 2B and $1e^{-4}$ for Mistral 7B and Meta Llama 3 70B across all tasks. We use Adam (Kingma, 2014) optimizer for Gemma 2B and Mistral 7B and AdaFactor (Shazeer & Stern, 2018) for Meta Llama 3 70B. We enable gradient checkpointing and use gradient accumulation with a mini batch size of 1 for all models.

**ILR.** We present the pseudo-code of ILR in Algorithm 1. Each round of ILR uses the same training configuration as the initial SFT, since the only difference is that the dataset is refined. When training the half-data models used for generating new proposals, we also adopt the same SFT training configuration for simplicity.

In our LM-simulated experiments, we only collect comparison feedback for proposals and initial demonstrations that are sufficiently different. In GSM8K and BIRD, this is determined by having different final numerical answer or execution result. For SaferPaca, we use OpenAI Embeddings API to compute the embeddings for the two responses, and then take dot product to compute their embedding distance. We only collect comparison feedback for pairs with the top 50% largest embedding distance.

**DPO.** We use the `DPOTrainer` in HuggingFace Transformers (Wolf et al., 2020) to perform DPO training. In all tasks, we use the suggested `rmsprop` optimizer with learning rate set to $1e^{-6}$, and we train for the same number of epochs as SFT. For all LM-simulated tasks, we sample 6 completion for each prompt, create 3 pairs with them, and then gather comparison feedback using $\tilde{q}$. We subsample the top 15% most confident feedback (measured by $|\tilde{q}(y_1 \succ y_2 \mid x) - 0.5|$) when constructing $\widetilde{\mathcal{D}}_{\text{RLHF}}$, since we find this performs better than using all feedback generated by the unreliable $\tilde{q}$ in DPO. Specifically, in GSM8K (2B $\rightarrow$ 7B), model accuracy after one round of DPO is 0.32±0.0043 when using all feedback and 0.36±0.0032 when using the 15% most confident feedback.

## D  HUMAN STUDY DETAILS

We use CloudResearch Connect[2] to recruit online workers to write human demonstrations for 1000 Alpaca (Taori et al., 2023) instructions and provide comparison feedback that are used in two rounds of ILR and DPO.

### D.1  COLLECTING HUMAN DEMONSTRATIONS

In our survey for collecting human demonstrations, we assign each worker 15 questions with 3 additional screening questions formatted in the same way for filtering out low-quality responses. Specifically, the screening questions and filtering critiria are:

1. **Instruction:** How many letter R's does the word 'STRAWBERRY' have?
   **Expected answer:** Anything containing 3 or three.

2. **Instruction:** Weng earns $12 an hour for babysitting. Yesterday, she just did 50 minutes of babysitting. How much did she earn?
   **Expected answer:** Anything containing 10 or ten.

3. **Instruction:** I want to know what dish I can cook with these ingredients: eggs, tomatoes, salt, oil. Also, please give me a list of steps to cook it.
   **Expected answer:** Anything containing the mentioned ingredients and at least 2 steps for cooking them.

The task instruction we provided at the beginning of the survey is:

> **Task instruction for writing response demonstrations**
>
> Thank you for participating in our study! We would like your help in training an AI assistant that can follow users' instructions in a helpful, honest, and harmless way.
>
> In this survey, you will be presented with 15 questions posed by human users to an AI chatbot. For each question, please take more than 1 minute and less than 2 minutes to write a response that you think is as helpful, honest, and harmless as possible. Please aim for high-quality detailed responses within the given time.
>
> Additionally, there will be 3 randomly placed quality check questions. These questions are super easy and you will definitely get them right if you are attentive. We will manually check your answers to

---

[2]https://connect.cloudresearch.com

these questions, and if you don't pass, your response may not be qualified. However, you will still receive a base payment of $0.75. In such a case, please manually return the assignment to ensure we can pay you the base amount. Please let us know if you find our judgement incorrect.

To get full compensation, please complete all questions. Otherwise, you will only receive a base payment of $0.75.

At the end, we welcome your feedback on improving the survey for future participants!

NOTE: Please refrain from searching for answers during the survey. No use of external sources (e.g., Google or ChatGPT) is allowed.

## D.2 COLLECTING HUMAN COMPARISON FEEDBACK

In each round of ILR and DPO, we collect human comparison feedback for all 1000 questions. For DPO, model completions are generated by models trained with $\beta = 0.1$. In our survey for collecting human comparison feedback, we assign each worker 50 questions with 3 additional screening questions formatted in the same way for filtering out low-quality responses. Specifically, the screening questions and filtering critiria are:

1. **Instruction:** Where is water that has its salt removed before it can be used as drinking water most likely to have come from?
   **Response A:** Water that has its salt removed before it can be used as drinking water is most likely to have come from a sea.
   **Response B:** Water that has its salt removed before it can be used as drinking water is most likely to have come from a lake.
   **Expected answer:** A with any confidence level.

2. **Instruction:** Help me solve this math problem. Input: How can I compute the area of a circle with radius 5?
   **Response A:** The area of it is $25\pi$.
   **Response B:** Note that the area of a circle with radius r is $\pi * r^2$. Therefore, the area of a circle with radius 5 is $\pi * 5 * 5 = 25\pi$.
   **Expected answer:** B with any confidence level.

3. **Instruction:** Immediately before and after running a 50 metre race, your pulse and breathing rates are taken. What changes would you expect to find?
   **Response A:** After running a 50-metre race, an increase in pulse and breathing rate is the changes one would expect to find immediately before and after.
   **Response B:** After running a 50 metre race, you would expect to find an increase in pulse but no change in breathing rate when your pulse and breathing rates are taken immediately before and after the race.
   **Expected answer:** A with any confidence level.

The task instruction we provided at the beginning of the survey is:

**Task instruction for providing comparison feedback**

Thank you so much for participating in our study! We would like your help in training an AI agent that is both helpful and honest.

First, you will be shown three screening questions, and if you get them correct, you will be able to participate in the rest of our survey. If you do not qualify, you will be eligible to receive a base payment of $0.75, and you will be automatically redirected to a Google Form. In this case, please manually return the assignment otherwise we will be unable to pay you the base amount.

For the main task, you will be shown fifty request-response groups: each request is posed by a user to an AI chatbot, and you will see two potential responses that the AI chatbot has generated. Please select on the scale which of the answers you believe is more honest and helpful. For instance, if you believe one of the choices is more helpful, select the bubble corresponding to one of the two

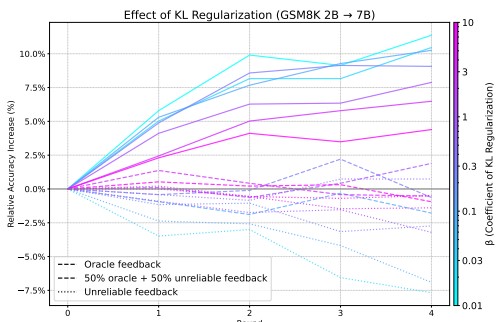 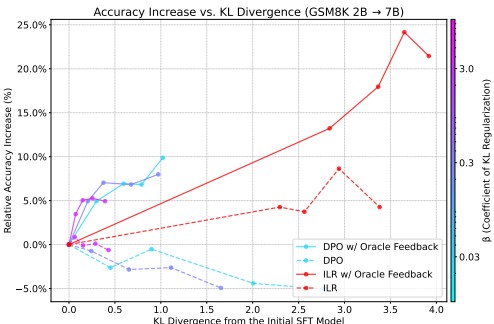

(a) DPO effectively improves SFT models with oracle feedback, especially with weak regularization. However, with unreliable feedback, improvements are limited to a narrow range of regularization strengths because weak regularization causes overoptimization.

(b) Strong regularization in DPO limits useful model updates, while small regularization leads to overoptimization of unreliable feedback. In contrast, ILR facilitates large model updates, allowing for faster improvement and efficient use of comparison feedback.

Figure 8: DPO struggles to avoid overoptimization of noisy preferences while also updating the initial suboptimal SFT model sufficiently to improve performance.

extremes on the scale, but if you believe the two choices are similar to each other, select a bubble in the center of the scale. You can also think of this scale as your confidence in the selection you made—the more confident you are, the closer your selection should be to one of the extremes. Optionally, you can explain why you chose the response that you did in the scratch space provided on each page.

You should spend around 30 seconds and no more than 1 minute for each question. The total time limit of this survey is 50 minutes. Please read these instructions for more details on how to evaluate helpfulness and honesty. You may open the instructions in a separte tab for reference whenever you want. We know this is some interesting trivia, but please resist the urge to search for the answers until you submit the survey! No use of external sources, including Google or ChatGPT, is permitted.

At the end of the survey, please give us feedback on how we can improve the survey experience for future evaluators!

NOTE: Please do not take this survey more than once! You will not be compensated for more than one attempt.

Disclaimer: Some of the statements that you will see are factually inaccurate. Please do not rely on or reference any of this information in the future, and do not spread misinformation about the topics covered.

### D.3 DPO AND ILR HYPEPERAMETERS

In both DPO and ILR, we only use comparison data with at least moderate confidence. We tested the first round of DPO using the top 10%, 20%, 30%, 40% and 80% most confident comparisons and found 30% yields the best performance, hence this is applied throughout the two rounds. We also tested $\beta \in \{0.01, 0.1, 0.5, 0.1\}$ for the first round and found $\beta = 0.1$ performs the best. For ILR, we tested $\alpha \in \{0.1, 0.2, 0.3\}$ for the first round and applied the best-performing $\alpha = 0.1$ across the two rounds.

## E ADDITIONAL EXPERIMENTS

### E.1 KL REGULARIZATION IN DPO

In Figure 8, we present the complete experiment results when using different levels of KL regularization and feedback quality for DPO.

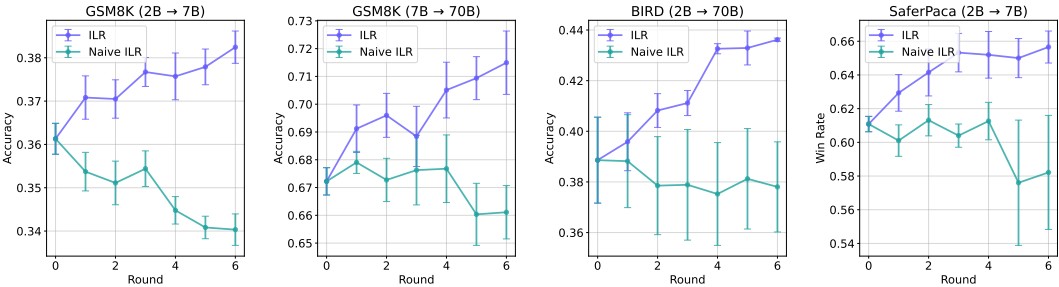

Figure 9: Naively replacing initial SFT labels with new model's proposals lead to performance degradation across all tasks, suggesting the importance of refinement oversight.

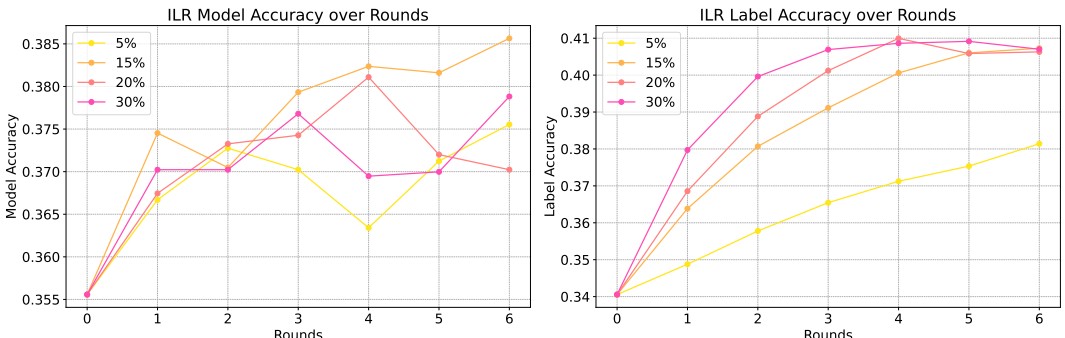

Figure 10: **Left:** Overall, ILR is not particularly sensitive to choice of $\alpha$, although larger $\alpha$ may lead to less stable performance, while small $\alpha$ does not allow enough improvement. **Right:** When $\alpha$ is large enough, SFT data label accuracy converge to a similar level. When $\alpha$ is overly small, label accuracy is not efficiently improved.

### E.2   FAILURE OF NAIVE ILR: SUPERVISON FOR REFINEMENT IS NECESSARY

To understand the importance of comparison feedback within ILR's refinement process, we compare it to a naive approach that directly replaces an $\alpha$ fraction of the original labels with new proposals without any feedback. As shown in Figure 9, this naive method leads to performance degradation in all tasks, showing that supervision in the refinement process, even if unreliable, is necessary. It could be that training on model outputs without any curation leads to model collapse, a phenomenon observed when training generative models with synthetic data (Shumailov et al., 2023; Ren et al., 2024; Gerstgrasser et al., 2024).

### E.3   EFFECT OF CONTROLLING REFINEMENT SPEED IN ILR

The refinement speed in ILR, controlled by $\alpha$, plays a crucial role in maintaining stability. Figure 10 illustrates the impact of different $\alpha$ values on model performance in GSM8K. We find $\alpha = 0.05$ too small for effective improvement of label accuracy, while large $\alpha$'s make the refinement process less stable. According to this result, we set $\alpha = 0.15$ for all other experiments in LM-based simulation and find that this choice leads to relatively stable performance across all tasks, showing that $\alpha$ is not particularly sensitive to change in data distribution. Future work can explore adaptive refinement speed to remove the potential need for manual hyperparameter tuning.

### E.4   IMPROVING DPO WITH ILR MODULES

Several modules in ILR can also be applied to DPO. In sDPO, we use two models trained on disjoint data subsets to generate samples for the preference dataset, similar to how we generate proposals with the cross-labeling framework in ILR. In wsDPO, we construct the preference dataset by comparing model-generated responses with original unreliable labels, similar to the mechanism of using feedback to decide label refinement in ILR. We test these two variants of DPO in the GSM8K (2B $\rightarrow$ 7B) setting. As shown in Table 2, these methods provide minimal improvements upon DPO. We conjecture

Table 2: Accuracies of DPO variants with ILR modules across 4 rounds. The highest value in each column is highlighted in bold.

| Method | Round 1 | Round 2 | Round 3 | Round 4 |
|--------|---------|---------|---------|---------|
| DPO | 0.3591 | 0.3520 | 0.3462 | 0.3412 |
| sDPO | 0.3624 | 0.3561 | 0.3457 | 0.3343 |
| wsDPO | 0.3626 | 0.3624 | 0.3538 | 0.3482 |
| ILR | **0.3647** | **0.3738** | **0.3851** | **0.3788** |

Table 3: Accuracies of DPO variants with improved loss functions across 4 rounds. The highest value in each column is highlighted in bold.

| Method | Round 1 | Round 2 | Round 3 | Round 4 |
|--------|---------|---------|---------|---------|
| DPO | 0.3591 | 0.3520 | 0.3462 | 0.3412 |
| IPO | 0.3604 | 0.3571 | 0.3503 | 0.3460 |
| cDPO | 0.3589 | 0.3536 | 0.3563 | 0.3530 |
| rDPO | 0.3568 | 0.3487 | 0.3386 | 0.3391 |
| ILR | **0.3647** | **0.3738** | **0.3851** | **0.3788** |

that they still struggles due to similar reasons to standard DPO: Errors learned during the initial SFT stage being hard to correct through preference optimization, and limitations caused by the overoptimization issues discussed in Section 4.1.

### E.5 ROBUST DPO LOSSES

Recent work has proposed several DPO losses that address preference noise. In Table 3, we compare IPO (Azar et al., 2024), cDPO (Mitchell, 2023), and rDPO (Chowdhury et al., 2024) to standard DPO and ILR. While these methods may help with random preference noise, they are less effective for handling unreliable supervision in our setting. This could be because both task demonstrations used in SFT *and* comparison feedback are unreliable in our setting. Moreover, noise in demonstrations or comparison feedback are systematic (Section 2) instead of random, unlike the random label flip assumed in some theoretical results (Mitchell, 2023; Chowdhury et al., 2024). Such systematic errors are particularly challenging because they may not be easily modeled using standard robust learning techniques.

