# OpenReview forum: "Iterative Label Refinement Matters More than Preference Optimization under Weak Supervision"
_ICLR.cc/2025/Conference — ICLR 2025 Spotlight_

### Official Review · Reviewer_mmnq · 2024-10-26

**Soundness:** 3
**Presentation:** 3
**Contribution:** 3
**Rating:** 8
**Confidence:** 2

**Summary:**

The paper shows that DPO is ineffective in the presence of unreliable feedback. The paper introduces a new method that replaces data samples in the SFT demonstrations dataset with synthetic demonstrations if the reward model prefers the synthetic demonstration. This process is iterated.

**Strengths:**

- Demonstrating the failure of DPO via overoptimization in the presence of unreliable feedback is important, given the prevalence of unreliable feedback.
- The introduced method is simple and performs clearly better.
- Validation on human setting increases confidence that the results on synthetic setting are reasonable

**Weaknesses:**

- Doesn't compare against PPO

**Questions:**

- Is it specifically necessary to do 1:1 replacement of training data points with new synthetic ones? Does it make sense to decouple the number of data points removed due to low quality, and the number of new synthetic data points accepted? How much of the effectiveness is due to the filtering vs the addition of synthetic data points?

---

> ### Author Response · Authors · 2024-11-20
> **Response to review (1/1)**
>
> We thank the reviewer for their thoughtful and review and for recognizing that “the introduced method is simple and performs clearly better.” We have addressed their points below.
>
> > Doesn't compare against PPO
> >
>
> Thank you for raising this important point about comparisons to PPO. While we focused on DPO since its stability and computational efficiency allowed us to conduct more thorough experiments, we have conducted additional experiments with several recent robust preference optimization methods on GSM8K (2B → 7B), which we will also include in the final paper:
>
> |  | Round 1 Acc | Round 2 Acc | Round 3 Acc | Round 4 Acc |
> | --- | --- | --- | --- | --- |
> | DPO | 0.3591 | 0.3520 | 0.3462 | 0.3412 |
> | IPO [1] | 0.3604 | 0.3571 | 0.3503 | 0.3460 |
> | cDPO [2] | 0.3589 | 0.3536 | 0.3563 | 0.3530 |
> | rDPO [3] | 0.3568 | 0.3487 | 0.3386 | 0.3391 |
> | ILR | **0.3647** | **0.3738** | **0.3851** | **0.3788** |
>
> All of these preference optimization methods show similar patterns under unreliable supervision. We agree that validating this with PPO experiments would strengthen our conclusions and plan to include this in future work.
>
> [1] Mitchell. "A note on DPO with noisy preferences & relationship to IPO." (2023).
>
> [2] Azar et al. "A general theoretical paradigm to understand learning from human preferences." (2024).
>
> [3] Chowdhury et al. "Provably robust dpo: Aligning language models with noisy feedback." (2024).
>
> > Is it specifically necessary to do 1:1 replacement of training data points with new synthetic ones? Does it make sense to decouple the number of data points removed due to low quality, and the number of new synthetic data points accepted? How much of the effectiveness is due to the filtering vs the addition of synthetic data points?
> >
>
> Thank you for this insightful question about ILR's refinement process. We conducted two ablation studies to understand the relative importance of removing low-quality examples versus adding synthetic ones. In the first ablation, we simply remove original labels rejected by the evaluator; in the second, we add chosen new proposals without removing old labels.
>
> |  | round 1 | round 2 | round 3 | round 4 | round 5 | round 6 |
> | --- | --- | --- | --- | --- | --- | --- |
> | Full ILR | **0.3745** | 0.3705 | 0.3793 | **0.3824** | **0.3816** | **0.3856** |
> | Only remove | 0.3647 | **0.3738** | **0.3851** | 0.3788 | 0.3813 | 0.3829 |
> | Only add | 0.3662 | 0.3682 | 0.3596 | 0.3700 | 0.3738 | 0.3707 |
>
> Surprisingly, our preliminary experiments on GSM8K (2B → 7B) show that:
>
> 1. Removing low quality labels identified during ILR provides most of ILR's benefits, achieving similar performance to full ILR.
> 2. Simply adding new data without removing old labels cannot recover full ILR’s performance.
>
> We hypothesize that proposals generated by the 7B model may not be high-quality enough to provide benefits beyond what we get from removing bad demonstrations. Also, pure removal strategies could be risky for tasks requiring coverage of diverse inputs such as pluralistic alignment [1]; the balance of removing and adding would help maintain both quality and coverage of the training data.
>
> To investigate these hypotheses, future work can conduct the ablations on larger models that might generate higher-quality synthetic data, and analyze ILR’s performance on tasks that require broad coverage of diverse inputs.
>
> [1] Sorensen et al. "A roadmap to pluralistic alignment." (2024).

---

### Official Review · Reviewer_bW5s · 2024-10-27

**Soundness:** 2
**Presentation:** 3
**Contribution:** 3
**Rating:** 8
**Confidence:** 3

**Summary:**

This paper proposes a novel approach, Iterative Label Refinement (ILR), aimed at improving language model (LM) post-training when demonstration and human supervision is unreliable. The authors identify that RLHF (specifically DPO) struggles under unreliable feedback due to overoptimization, and they propose ILR as an alternative that improves the supervised fine-tuning (SFT) dataset itself. The approach replaces low-quality human demonstrations with (small) model-generated alternatives based on comparison feedback, which is then used to retrain the model. The paper provides evidence that ILR outperforms DPO in various tasks, including math, coding and safe instruction-following, under both LM-simulated and human supervision settings.

Comments:
- Scalability and Computational Cost: The paper could include a more detailed discussion on the computational trade-offs involved in using ILR versus DPO. Given that ILR requires multiple rounds of refinement, it would be useful to know how this affects training time and computational resources.
- Clarification on Theoretical Boundaries: Theoretical analysis could be expanded to explore the long-term behavior of ILR. For example, will the refinement process converge effectively without introducing new errors or biases into the dataset over time?
- Additional Visualizations: It would be helpful to see additional visualizations that show the progression of ILR versus DPO over multiple training rounds, especially in terms of how the refined dataset improves in quality and how the model’s performance evolves.
- Task Extension: Future work could explore extending ILR to more diverse task types, including more subjective ones or tasks where human demonstrations are more error-prone (e.g., complex visual tasks or tasks requiring nuanced judgment).

**Strengths:**

- Novelty and Significance: ILR introduces a fresh approach to addressing the key challenge of unreliable human supervision in language model post-training. This is quite crucial as RLHF is becoming increasingly important for scaling LMs in practical applications.
- Comprehensive Evaluation: The experiments cover different tasks including math, coding, safe instruction-following and use both LM and real human feedback, ensuring the findings are well-supported. The robustness of ILR under different conditions and tasks is thoroughly tested and clearly explained.
- Clear Motivation: The paper presents a strong investigation and reasoning for why RLHF, especially DPO, struggles under weak supervision. The overoptimization problem is well-identified and investigated, which motivates the need for a method like ILR, which focuses on refining the training data rather than the model directly.
- Effective Use of Feedback: The authors highlight the importance of directing comparison feedback towards improving the dataset, not just the model. This is a significant insight and could influence future work in RLHF, encouraging researchers to rethink more about how to utilize human feedback in post-training.

**Weaknesses:**

- Potential ensembling effect that may explain performance gain: 5.1 describes training one model on each half of the SFT data, then using these models to cross-label during ILR. There does not seem to be an ablation either. I think a fair comparison would be against a two-model ensemble, as opposed to a single model baseline.
- Limited Comparison with other RLHF Methods: While the paper focuses on comparing ILR to DPO, it does not address how ILR might perform compared to other RLHF algorithms, like PPO. Including more comparisons could make the paper’s conclusions more robust and introduce a broader impact.
- Computational Cost and Scalability: The ILR approach, which requires iterative refinement and cross-evaluation of datasets, which means  both SFT model and DPO needs to be retrained for every iteration. This might introduce higher computational complexity compared to DPO or traditional RLHF methods. The paper could benefit from a more detailed discussion of the trade-offs between performance improvement and computational cost.
- Task Diversity: While the paper tests ILR on math, coding, and safe instruction-following, it would be beneficial to see it applied to a broader range of tasks, particularly more complex or open-ended ones where human feedback is highly subjective or unreliable (e.g., creative writing or subjective question-answering or toxicity).
- Theoretical Depth: The theoretical tracking of ILR is solid, but the paper could elaborate more on the theoretical limits of ILR. In particular, the long-term stability of ILR, and whether it risks introducing new forms of bias through the iterative refinement process, could be explored.

**Questions:**

- How does ILR compare to other RLHF methods like PPO? Would similar overoptimization issues arise in other preference optimization methods, or are they more resistant to unreliable supervision?
- Could ILR be extended or adapted to address tasks that involve more subjective or open-ended feedback, such as creative writing or opinion-based tasks?
- Does ILR introduce any risks of bias through its iterative process, particularly as models continue to generate and refine their own data? How would you address or mitigate such risks?

---

> ### Author Response · Authors · 2024-11-20
> **Response to review (1/2)**
>
> We thank the reviewer for their insightful review. We appreciate that they found our approach is “fresh”, “thoroughly tested and clearly explained”, and “encourages researchers to rethink more about how to utilize human feedback in post-training.” We have responded to their individual points below.
>
> > Potential ensembling effect that may explain performance gain: 5.1 describes training one model on each half of the SFT data, then using these models to cross-label during ILR. There does not seem to be an ablation either. I think a fair comparison would be against a two-model ensemble, as opposed to a single model baseline.
> >
>
> Thank you for raising this important concern. We did test a version of DPO that uses two models trained on disjoint data subsets to generate proposals in our preliminary ablation studies. Results on GSM8K (2B → 7B) show this method (denoted as sDPO) provides minimal benefit:
>
> |  | Round 1 Acc | Round 2 Acc | Round 3 Acc | Round 4 Acc |
> | --- | --- | --- | --- | --- |
> | DPO | 0.3591 | 0.3520 | 0.3462 | 0.3412 |
> | sDPO | 0.3624 | 0.3561 | 0.3457 | 0.3343 |
> | ILR  | **0.3647** | **0.3738** | **0.3851** | **0.3788** |
>
> We conjecture that even with potentially better proposals from two models, sDPO still struggles due to similar reasons to standard DPO. It suffers from 1) difficulty correcting errors learned during initial SFT during preference optimization and 2) overoptimization issues caused by unreliable feedback, as discussed in Section 4.1.
>
> > Limited Comparison with other RLHF Methods: While the paper focuses on comparing ILR to DPO, it does not address how ILR might perform compared to other RLHF algorithms, like PPO. Including more comparisons could make the paper’s conclusions more robust and introduce a broader impact.
> >
>
> Thank you for highlighting the need for broader comparisons. We've evaluated several recent algorithms designed to handle noisy preferences on GSM8K (2B → 7B) and will include these comparisons in the final paper. As shown below, they all perform similar to DPO and underperform ILR:
>
> |  | Round 1 Acc | Round 2 Acc | Round 3 Acc | Round 4 Acc |
> | --- | --- | --- | --- | --- |
> | DPO | 0.3591 | 0.3520 | 0.3462 | 0.3412 |
> | IPO [1] | 0.3604 | 0.3571 | 0.3503 | 0.3460 |
> | cDPO [2] | 0.3589 | 0.3536 | 0.3563 | 0.3530 |
> | rDPO [3] | 0.3568 | 0.3487 | 0.3386 | 0.3391 |
> | ILR | **0.3647** | **0.3738** | **0.3851** | **0.3788** |
>
> Our work is focused on DPO-based methods due to their stability and computational efficiency, which enabled our thorough experiments using limited compute. Also, the above results suggest that preference optimization methods face similar challenges under unreliable supervision. Nonetheless, we agree that extending these findings to PPO would be valuable future work.
>
> [1] Mitchell. "A note on DPO with noisy preferences & relationship to IPO." (2023).
>
> [2] Azar et al. "A general theoretical paradigm to understand learning from human preferences." (2024).
>
> [3] Chowdhury et al. "Provably robust dpo: Aligning language models with noisy feedback." (2024).
>
> > Computational Cost and Scalability: The ILR approach, which requires iterative refinement and cross-evaluation of datasets, which means both SFT model and DPO needs to be retrained for every iteration. This might introduce higher computational complexity compared to DPO or traditional RLHF methods. The paper could benefit from a more detailed discussion of the trade-offs between performance improvement and computational cost.
> >
>
> Thank you for raising this practical consideration. We’d like to clarify a few points:
>
> 1. ILR does not involve DPO training—it only requires SFT in every iteration.
> 2. ILR trains one SFT model plus two additional half-data models, requiring about 2N forward passes where N is the number of training samples.
> 3. DPO only trains one model in each round, but it uses both a chosen and a rejected response for each prompt; moreover, it requires computing logits of the reference model. Therefore, it requires about 3N forward passes, which is actually more inefficient than ILR.
> 4. Nonetheless, compared to cost of pretraining, the difference in the costs of ILR and DPO should be almost negligible.

---

> ### Author Response · Authors · 2024-11-20
> **Response to review (2/2)**
>
> > Task Diversity: While the paper tests ILR on math, coding, and safe instruction-following, it would be beneficial to see it applied to a broader range of tasks, particularly more complex or open-ended ones where human feedback is highly subjective or unreliable (e.g., creative writing or subjective question-answering or toxicity).
> >
>
> Thank you for this suggestion. We note that the SaferPaca dataset we use already includes subjective tasks about following instructions (which includes question-answering) and rejecting unsafe or toxic prompts. That said, we agree that exploring additional complex domains could further demonstrate the robustness of our results.
>
> > Theoretical Depth: The theoretical tracking of ILR is solid, but the paper could elaborate more on the theoretical limits of ILR. In particular, the long-term stability of ILR, and whether it risks introducing new forms of bias through the iterative refinement process, could be explored.
> >
>
> We appreciate the request for deeper theoretical analysis. Our main goal in this work is to provide empirical evidence that current methods may break down under unreliable supervision, and that an alternative method could work. We agree that understanding ILR's theoretical properties is crucial, and we aim to explore it in our future work.
>
> > Does ILR introduce any risks of bias through its iterative process, particularly as models continue to generate and refine their own data? How would you address or mitigate such risks?
> >
>
> In practice, ILR would be deployed with human oversight—similar to our time-constrained human study—rather than automatically replacing human labeled data with synthetic data. Our experiments with naive ILR (Section 5.2) show that simply replacing demonstrations with model outputs without supervision leads to poor results, indicating that incorporating human oversight can reduce risks during the iterative process. We believe that these risks could be further mitigated when combining ILR with other scalable oversight techniques, such as decomposing LM-generated responses [1] or using other LM to discover flaws in the synthetic data [2].
>
> [1] Wen et al. “Learning Task Decomposition to Assist Humans in Competitive Programming.” (2024)
>
> [2] McAleese et al. “LLM Critics Help Catch LLM Bugs.” (2024)

---

> ### Comment · Reviewer_bW5s · 2024-11-20
>
> Thank you authors for your detailed response + additional experiments. I have increased my score and advocate for acceptance!

---

### Official Review · Reviewer_J4qi · 2024-10-27

**Soundness:** 2
**Presentation:** 3
**Contribution:** 2
**Rating:** 5
**Confidence:** 4

**Summary:**

This paper proposes iterative label refinement (ILR) as an alternative to reinforcement learning from human feedback (RLHF) for language model post-training under unreliable supervision. The authors argue that in settings where human supervision is unreliable, methods such as direct preference optimization (DPO) become less effective. Instead, ILR iteratively refines the supervised fine-tuning (SFT) dataset by replacing unreliable human demonstrations with better model-generated responses, thus improving the model’s training data. Experiments show that ILR outperforms DPO in tasks such as math, coding, and safe instruction-following, especially when human supervision is noisy or unreliable.

**Strengths:**

1. **Motivation.** Safe post-training of language models with noisy fine-tuning data is a highly practical problem with a lot of relevant research done to address the issue.

2. **Experiments and analyses.** The paper conducts a range of experiments, simulating unreliable supervision using small LMs and time-constrained human evaluators to evaluate ILR under multiple settings.

3. **Performance.** ILR demonstrates improvements over DPO under the unreliable supervision settings considered in the work.

**Weaknesses:**

1. **Limited evaluation of alternatives.** Several approaches exist for handling noisy feedback data, including conservative DPO [1], which combines regular DPO with label smoothing, IPO [2], MMPO [3], and filtering noisy labels, among others. However, empirical evaluations of some of these methods in the unreliable supervision setting considered are missing.

2. **Unreliable supervision not well-defined.** The concept of unreliability is not quantitatively well-defined. For example, it is unclear whether a small LM trained as a classification model actually provides unreliable comparison feedback, and if so, to what extent the labels are considered inaccurate. Relatedly, it is unclear how unreliable the “time-constrained” human supervision really is.

3. **Questionable assumptions.** For the ILR framework to be effective, it seems that the SFT model needs to generate responses that are often better than those in the SFT dataset, and the annotator must be able to identify these better responses. While the latter assumption is discussed in the paper, the former depends on the SFT model being sufficiently capable and not overfitted to the SFT data, so that it can generate responses than the SFT data at least some of the time. This depends on factors such as the size of the SFT dataset, the proportion of unreliable data, and the size of the SFT model, raising questions about how generally applicable ILR is.

---
[1] Mitchell. "A note on DPO with noisy preferences & relationship to IPO." (2023). \
[2] Azar et al. "A general theoretical paradigm to understand learning from human preferences." (2024). \
[3] Kim et al. “Margin Matching Preference Optimization: Enhanced Model Alignment with Granular Feedback.” (2024).

**Questions:**

1. How does ILR compare to having annotators review each sample in the SFT data to remove low-quality ones? This approach seems less expensive, as it avoids generating new responses with the SFT model and lets the annotators avoid pairwise comparisons.

2. How frequently are the model-generated responses considered better than the SFT data depending on the size of the model?

3. Typos: “Ground Truth” in Figure 5.

---

> ### Author Response · Authors · 2024-11-20
> **Response to review (1/2)**
>
> We thank the reviewer for their detailed and thoughtful comments. We appreciate their recognition that our work addresses a “highly practical problem”. We have addressed their specific concerns below.
>
> > **Limited evaluation of alternatives.** Several approaches exist for handling noisy feedback data, including conservative DPO [1], which combines regular DPO with label smoothing, IPO [2], MMPO [3], and filtering noisy labels, among others. However, empirical evaluations of some of these methods in the unreliable supervision setting considered are missing.
> >
>
> Thank you for highlighting these important baselines. We have conducted additional experiments with conservative DPO (cDPO) [1], IPO [2], and robust DPO (rDPO) [3] on GSM8K (2B → 7B): (we will include these results in the final paper)
>
> |  | Round 1 Acc | Round 2 Acc | Round 3 Acc | Round 4 Acc |
> | --- | --- | --- | --- | --- |
> | DPO | 0.3591 | 0.3520 | 0.3462 | 0.3412 |
> | IPO [1] | 0.3604 | 0.3571 | 0.3503 | 0.3460 |
> | cDPO [2] | 0.3589 | 0.3536 | 0.3563 | 0.3530 |
> | rDPO [3] | 0.3568 | 0.3487 | 0.3386 | 0.3391 |
> | ILR | **0.3647** | **0.3738** | **0.3851** | **0.3788** |
>
> These results demonstrate that while these methods may help with random preference noise, they are less effective for handling unreliable supervision in our setting. This could be because 1) in our setting, both task demonstrations used in SFT and comparison feedback used in DPO are unreliable, and 2) noise in demonstrations or comparison feedback are systematic (Section 2) instead of random, unlike the random label flip assumed in some theoretical results [1, 3]. Such systematic errors are particularly challenging because they may not be easily modeled using standard robust learning techniques.
>
> We haven’t included MMPO [4] in our experiments since it is not explicitly designed to address noise and it is not implemented in the HuggingFace framework that our experiment pipeline is based on. We will further explore this method in future work.
>
> [1] Mitchell. "A note on DPO with noisy preferences & relationship to IPO." (2023).
>
> [2] Azar et al. "A general theoretical paradigm to understand learning from human preferences." (2024).
>
> [3] Chowdhury et al. "Provably robust dpo: Aligning language models with noisy feedback." (2024).
>
> [4] Kim et al. “Margin Matching Preference Optimization: Enhanced Model Alignment with Granular Feedback.” (2024).
>
> > The concept of unreliability is not quantitatively well-defined. For example, it is unclear whether a small LM trained as a classification model actually provides unreliable comparison feedback, and if so, to what extent the labels are considered inaccurate. Relatedly, it is unclear how unreliable the “time-constrained” human supervision really is.
> >
>
> Thank you for raising this important concern about quantifying unreliability.
>
> Firstly, we can measure unreliability using task accuracy in our math (GSM8K) and coding (BIRD). For example, in GSM8K, unreliability of task demonstrations can be measured by comparing them with ground truth. Figure 5a shows that LM-generated demonstrations for GSM8K (2B → 7B) only has an accuracy of 0.34, which can be considered unreliable.
>
> For comparison feedback, we measure how often the evaluator selects the objectively better answer (GSM8K 2B → 8B):
>
> |  | round 1 | round 2 | round 3 | round 4 | round 5 | Round 6 |
> | --- | --- | --- | --- | --- | --- | --- |
> | correct replacement rate | 0.8565 | 0.7347 | 0.6568 | 0.6502 | 0.5822 | 0.5141. |
>
> This declining accuracy over rounds demonstrates the evaluator's increasing difficulty in identifying better responses. Moreover, it is shown in Figure 5 that an unreliable evaluator can not improve the SFT data to a level that can be achieved with an oracle.
>
> For our human studies, we also have multiple indicators of unreliability: 1) Response length: Figure 6 shows time-constrained human responses are significantly shorter than GPT-4 responses; 2) Qualitative analysis: Figure 7 demonstrates clear misconceptions in human responses; 3) No qualification requirements: We recruited participants without domain expertise requirements through CloudResearch Connect.
>
> While measuring unreliability in more subjective tasks like SaferPaca is challenging, our primary goal is to demonstrate that when supervision quality falls below a certain threshold, existing methods break down and alternatives like ILR become necessary.
>
> Nonetheless, we believe that more nuanced understanding of how different level of unreliability affects ILR and RLHF would be an interesting direction that could be explored in future work.

---

> ### Author Response · Authors · 2024-11-20
> **Response to review (2/2)**
>
> > For the ILR framework to be effective, it seems that the SFT model needs to generate responses that are often better than those in the SFT dataset, and the annotator must be able to identify these better responses. While the latter assumption is discussed in the paper, the former depends on the SFT model being sufficiently capable and not overfitted to the SFT data, so that it can generate responses than the SFT data at least some of the time. This depends on factors such as the size of the SFT dataset, the proportion of unreliable data, and the size of the SFT model, raising questions about how generally applicable ILR is.
> >
>
> Thank you for this important point about ILR's prerequisites. However, we note that RLHF methods also rely on a similar assumption—that models can generate some responses better what humans can provide [1]. This is fundamental to the entire paradigm of improving models through feedback rather than direct demonstration.
>
> The phenomenon of models sometimes outperforming their training data, while not yet theoretically guaranteed, is an empirical fact observed across all tasks and model sizes considered in our work (Section 4) as well as recent work [2, 3]. Moreover, Figure 7 provides a concrete example where the SFT model generates better responses than time-constrained human annotators.
>
> [1] Stiennon et al. “Learning to summarize from human feedback.” (2020)
>
> [2] Burns et al. “Weak-to-Strong Generalization: Eliciting Strong Capabilities With Weak Supervision.” (2023)
>
> [3] Guo et al. “Vision Superalignment: Weak-to-Strong Generalization for Vision Foundation Models.” (2024)
>
> > How does ILR compare to having annotators review each sample in the SFT data to remove low-quality ones? This approach seems less expensive, as it avoids generating new responses with the SFT model and lets the annotators avoid pairwise comparisons.
> >
>
> We agree that data filtering in general helps with learning under weak supervision. Asking annotators to think twice, review their answers, and rewrite or remove should improve data quality, but it seems orthogonal to our assumption that the final demonstration data we obtain is unreliable. As an evidence, some of ChatGPT’s training data initially rated as “flawless” has only been found in recent work [1] to contain flaws.
>
> In addition, pairwise comparisons are often cognitively easier than absolute quality judgments. For example, while an annotator might struggle to identify a mistake in a mathematical solution simply by proofreading, they may be able to notice the it when a correct solution is provided—in this case the mistake will be revealed by the difference between the two solutions.
>
> [1] McAleese et al. “LLM Critics Help Catch LLM Bugs.” (2024)
>
> > How frequently are the model-generated responses considered better than the SFT data depending on the size of the model?
> >
>
> Here's how often model-generated proposals were accepted across rounds in two settings of GSM8K:
>
> |  | round 1 | round 2 | round 3 | round 4 | round 5 | round 6 |
> | --- | --- | --- | --- | --- | --- | --- |
> | GSM8K 2B → 7B | 0.2684 | 0.2269 | 0.2074 | 0.1670 | 0.1373 | 0.1343 |
> | GSM8K 7B → 70B | 0.1897 | 0.1507 | 0.1284 | 0.1003 | 0.0937 | 0.0733 |
>
> Higher acceptance rates in the 2B → 7B setting suggest that there could be more room for improvement when initial demonstrations are more unreliable (as those generated by the 2B model are, compared to those generated by the 7B model).
>
> Additionally, declining acceptance rates across rounds in both settings indicate the increasing difficulty of obtaining further improvements as data quality increases. This could be due to either 1) the evaluator struggling to identify better responses, since those that can be identified are already used or 2) the model is generating less responses that are better than the original demonstrations.
>
> Nonetheless, we acknowledge that a more systematic study controlling for either supervisor or student model size while varying the other (e.g., compare 2B → 7B and 2B → 70B) would be valuable future work.
>
> > Typos: “Ground Truth” in Figure 5.
> >
>
> Thank you for catching this typo! We will fix it in our revision.

---

> ### Author Response · Authors · 2024-11-29
>
> Dear reviewer J4qi,
>
> We have added new experiments and explanations to address your concerns in our rebuttal. As the discussion period is concluding soon, please let us know if there are any additional questions we can address. Thank you!
>
> Best,
> Authors

---

### Official Review · Reviewer_R8qz · 2024-10-28

**Soundness:** 3
**Presentation:** 3
**Contribution:** 3
**Rating:** 8
**Confidence:** 4

**Summary:**

The authors examine how to improve model performance when SFT and preference data is unreliable (contains some proportion of mistakes). They find that doing SFT training still results in improvements with unreliable data, but DPO does not improve. They propose a new method, iterative label refinement, which involves iteratively training models on data subsets, and then using the models to relabel the SFT data (with an unreliable supervisor deciding if the new label should be accepted), and retraining. The proposed method outperforms DPO when applied to SFT models both in an artificial setup (using small models to train larger ones) and a time-constrained human-labeller setup.

**Strengths:**

- I think the experimental setup is good, looking at a variety of models and tasks, and the human labeling experiment is also interesting.
- The analysis of why DPO does not perform well with unreliable data is interesting, and provides reasonable evidence that DPO can fit too hard to the unreliable preferences.
- The performance of the proposed method seems robust, with it being tested both in multiple settings and with (time-constrained) human annotators. The performance gains seem robust across the tasks and models tested.

**Weaknesses:**

- Missing Baseline: it would be good to compare against just doing SFT on the chosen sample or unreliable DPO pairs (my understanding is that this is not the same as training only on the model’s proposals because there is an additional trained classifier that may pick the right answer, albeit unreliably).
- I wonder if the comparison between DPO and SFT/ILR methods are entirely fair since the DPO method doesn’t use the original ground truth answer, just two model generations. Could the DPO results perform better if the original ground truth label was also provided as an option when picking chosen and rejected pairs (as it is for the ILR approach)? Would it be possible to run the same ILR algorithm outlined in section 5.1, but rather than replace labels, just use the accepted/rejected proposals to construct chosen/rejected pairs instead?
- Figure 6b is quite hard to parse, as the label box overlaps with a large chunk of the figure, and the differences between round 1 and 2 are somewhat small.
- Did the authors try longer DPO training with a high beta? They clearly show that using a larger beta with unreliable feedback can improve a bit, I wonder if training for longer with higher beta results in further improvements with DPO or if it then starts to over-optimize as well (potentially a different way that the issues with DPO could be alleviated).

Overall, I think the paper presents an interesting issue (that DPO doesn't seem to work well with unreliable label) and proposes an interesting alternative method that appears to work well in across a few models and tasks, although I think the comparisons between DPO and ILR could be made a bit tighter.

**Questions:**

1. I was a bit unclear on how the unreliable DPO dataset is constructed: so, you generate responses from a model checkpoint, and train a classifier to pick the original ground truth label instead of the model response, and then use this classifier to pick between two samples from the same model checkpoint? Or is the classifier the same across rounds?

---

> ### Author Response · Authors · 2024-11-20
> **Response to review (1/2)**
>
> We thank the reviewer for their insightful review and suggestions. We are glad they found that “the experimental setup is good” and our human experiments and analysis of DPO are “interesting”. We have responded to their individual points below.
>
> > Missing Baseline: it would be good to compare against just doing SFT on the chosen sample or unreliable DPO pairs.
> >
>
> Thank you for suggesting this important baseline. We tested this in our preliminary ablation study, but we found that it led to immediate performance degradation in the fist round. On GSM8K (2B → 7B), after just 1 round of SFT on chosen samples generated by the initial SFT model, accuracy decreases from 0.3556 to 0.3497 (averaged over 3 seeds). This highlights a key insight behind ILR's design: it's crucial to maintain the full dataset to include originally high-quality demonstrations while replacing low-quality ones with new ”chosen” proposals, because simply training on “chosen” samples may lose valuable information present in the original dataset.
>
> > I wonder if the comparison between DPO and SFT/ILR methods are entirely fair since the DPO method doesn’t use the original ground truth answer, just two model generations. Could the DPO results perform better if the original ground truth label was also provided as an option when picking chosen and rejected pairs (as it is for the ILR approach)?
> >
>
> Thank you for this important question. To clarify: neither ILR nor DPO use ground truth labels in our experiments—both methods rely solely on unreliable supervision. We did explore a variant of DPO (which we call wsDPO) that constructs preference pairs by comparing model proposals against original demonstrations rather than only comparing between model proposals. While this seemed to slightly improve over standard DPO, it still underperformed ILR. For example, in GSM8K (2B → 7B):
>
> |  | Round 1 Acc | Round 2 Acc | Round 3 Acc | Round 4 Acc |
> | --- | --- | --- | --- | --- |
> | DPO | 0.3591 | 0.3520 | 0.3462 | 0.3412 |
> | wsDPO | 0.3626 | 0.3624 | 0.3538 | 0.3482 |
> | ILR | **0.3647** | **0.3738** | **0.3851** | **0.3788** |
>
> This underperformance could be due to errors learned during the initial SFT stage being hard to correct through preference optimization, similar to DPO. Also, wsDPO still suffers from the overoptimization issues discussed in Section 4.1 under unreliable feedback.
>
> > Would it be possible to run the same ILR algorithm outlined in section 5.1, but rather than replace labels, just use the accepted/rejected proposals to construct chosen/rejected pairs instead?
> >
>
> Thank you for this suggestion. We intentionally designed ILR to avoid using chosen/rejected pairs for preference optimization, as our analysis in Section 4.1 showed that preference optimization methods tend to suffer from overoptimization under unreliable supervision. However, we agree that finding ways to penalize poor responses could be valuable in addition to making use of good model-generated proposals. In future work, we plan to explore algorithms that can leverage both positive and negative feedback while remaining robust to unreliable supervision.
>
> > Figure 6b is quite hard to parse, as the label box overlaps with a large chunk of the figure, and the differences between round 1 and 2 are somewhat small.
> >
>
> Thank you for pointing out this problem! We will adjust the figure to make it clearer in our final paper.

---

> ### Author Response · Authors · 2024-11-20
> **Response to review (2/2)**
>
> > Did the authors try longer DPO training with a high beta? They clearly show that using a larger beta with unreliable feedback can improve a bit, I wonder if training for longer with higher beta results in further improvements with DPO or if it then starts to over-optimize as well.
> >
>
> As shown in Figure 3a, even with 4 rounds of training, high-beta DPO shows minimal improvement when using partially unreliable feedback (50% unreliable) and even less with fully unreliable feedback. While longer training with high beta might yield marginal gains, this comes at significant computational cost. Furthermore, new preference comparisons need to be collected for each round of DPO, so it would be expensive in terms of annotator time. More importantly, the fundamental limitation that strong regularization constrains model updates still persist, making it difficult to correct errors learned during the initial SFT stage. ILR avoids this trade-off by improving the training data itself rather than attempting to optimize unreliable preferences directly, and it does not require longer training time.
>
> > I was a bit unclear on how the unreliable DPO dataset is constructed: so, you generate responses from a model checkpoint, and train a classifier to pick the original ground truth label instead of the model response, and then use this classifier to pick between two samples from the same model checkpoint? Or is the classifier the same across rounds?
> >
>
> Yes, the classifier is the same across rounds and it is the same as the classifier in ILR. This ensures both methods operate under identical unreliable supervision conditions. More training details of this classifier can be found in Appendix B.

---

> > ### Comment · Reviewer_R8qz · 2024-11-24
> > **Response**
> >
> > Hi, thank you for your detailed and clarifying response - the notes about the chosen-only baseline and wsDPO experiments are quite useful. I've read through the other reviews and responses and am raising my score - I think the additional results shown make the results in this paper more solid and interesting.

---

### Author Response · Authors · 2024-11-27
**Paper revision**

We would like to express our gratitude to the reviewers for their insightful feedback and thorough reviews. We have carefully addressed their concerns in our rebuttals and have just uploaded a revised version of our paper, which includes the following updates:

- In Appendix E, we included experimental results for several baselines mentioned by Reviewer J4qi and the variants of DPO mentioned by Reviewers R8qz and bW5s.
- We corrected the typo in Figure 5 and revised the layout of Figure 6 according to the feedback from Reviewer J4qi and R8qz.
- We have updated the abstract to enhance conciseness and improved the overall clarity of the main text.

As the discussion period is coming to an end, please let us know if there are any additional questions or concerns we can address.

---

### Meta-Review · Area_Chair_CcGt · 2024-12-20

**Metareview:**

The paper proposes a new method for aligning LLMs specifically when SFT and preference data is unreliable (contains some proportion of mistakes). They propose a new method, iterative label refinement, which involves iteratively training models on subsets of supervised data and using the models to relabel the SFT data and retraining on it. The proposed method outperforms baselines like PPO and DPO when applied to SFT models in math, coding and safe instruction-following, under both LM-simulated and human supervision settings.

After the discussion period, 3 reviewers voted to accept and 1 reviewer remained marginal reject. The reviewers all agreed that safe post-training of LLMs, especially with noisy fine-tuning data is a highly practical problem, that the paper conducts sufficient experiments and analysis, and that the proposed method achieves strong performance especially under the unreliable supervision settings considered in the paper.

There were some concerns at the beginning around lack of comparisons to PPO and other RLHF baselines, as well as some missing ablations and analysis of key design decisions in the method. The authors presented an extensive rebuttal and addressed most of these concerns, and most of the reviewers were satisfied with these changes. I side with the majority of reviewers and agree that this is a solid contribution with strong results and analysis, and I believe that it meets the bar for acceptance.

**Additional Comments On Reviewer Discussion:**

There were some concerns at the beginning around lack of comparisons to PPO and other RLHF baselines, as well as some missing ablations and analysis of key design decisions in the method. The authors presented an extensive rebuttal and addressed most of these concerns, and several reviewers raised their scores to accept, good paper.

Reviewer J4qi remained marginal reject raised similar concerns wrt missing baselines, the definition and setup of 'unreliable supervision', and the analysis of key design decisions in the method specifically that the SFT model needs to generate responses that are often better than those in the SFT dataset. Although the reviewer did not raise their scores and acknowledge the rebuttal, I believe the authors have addressed all these points, including running the additional baselines requested, quantifying their proposed definition of unreliability, and running the analysis of small vs large model. So I am happy with the author's work and feel that it has adequately addressed the concerns.

---

### Decision · Program_Chairs · 2025-01-22

Accept (Spotlight)